# Recent Advances in the Remediation of Textile-Dye-Containing Wastewater: Prioritizing Human Health and Sustainable Wastewater Treatment

Aravin Prince Periyasamy 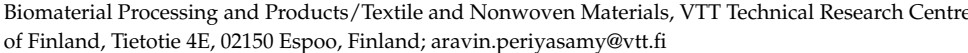

Biomaterial Processing and Products/Textile and Nonwoven Materials, VTT Technical Research Centre of Finland, Tietotie 4E, 02150 Espoo, Finland; aravin.periyasamy@vtt.fi

**Abstract:** Water makes up most of the Earth, although just 0.3% is usable for people and animals. The huge oceans, icecaps, and other non-potable water resources make up the remaining 99.7%. Water quality has declined in recent decades due to pollution from population growth, industry, unplanned urbanization, and poor water management. The textile industry has significant global importance, although it also stands as a major contributor to wastewater generation, leading to water depletion and ecotoxicity. This issue arises from the extensive utilization of harmful chemicals, notably dyes. The main aim of this review article is to combine and assess the impacts of textile wastewater that contains dyes and chemicals, and to examine their potential consequences on human health, aquatic health, and the environment. Moreover, the dedicated section presents an in-depth review of various environmentally sustainable approaches for the management and treatment of wastewater in the textile industry. These approaches encompass bio adsorbents, biological methods, membrane technology, ion exchange, advanced oxidation processes, as well as physicochemical and biochemical processes. Furthermore, this study also evaluates the contemporary progressions in this particular domain, taking into account the corresponding advantages and disadvantages. Finally, this article highlights the significance of recovering and reusing dyes, alkalis, and electrolytes in wastewater treatment. Additionally, it emphasizes the necessity of performing technoeconomic analyses and life cycle assessments (LCA) on wastewater treatment plants.

**Keywords:** effluents; ecotoxicity; dyes; carcinogenicity; textile wastewater; sustainability

---




## 1. Introduction

Currently, ecosystems are primarily experiencing harm due to the exhaustion of natural resources and the deterioration of the environment resulting from industrial expansion and environmental emergencies [1,2]. Water pollution is a significant environmental issue posing significant risks to water, the primary life-sustaining element on Earth, emphasizing its crucial role in supporting life [3,4]. Pollution is primarily caused by the insufficient potable water supply and the harmful exposure to various chemicals and pathogens in the polluted water and food chain. Water pollution is largely defined by two main problems: the lack of safe drinking water and the dangerous exposure to various chemicals and pathogens found in contaminated water and the food chain [5,6]. Water pollution is characterized by the overabundance of harmful substances in water bodies, resulting from both natural and human activities [7,8].

The textile industry is a significant contributor to water pollution [9], and it is also responsible for approximately 20% of global water pollution [10], as the second largest polluter after the oil industry [10]. In comparison to other industrial sectors, the textile industry is known to have the highest water and chemical consumption, with over 8000 species being utilized [11–13]. The wastewater generated by this industry is often characterized by a significant amount of unfixed colors and dyeing auxiliaries [14–17]. Approximately 800,000 tons of dyes are produced annually, with 10–15% of this quantity being lost to the

environment [18]. Over 10,000 distinct synthetic dye varieties have been introduced, with 70% of them belonging to the azo type. In general, dyes are classified into various types such as direct, reactive, basic, acidic, disperse, vat, sulfur, metal complex, and mordant dyes [10].

Dyes are a class of organic compounds that possess the ability to impart color to a diverse range of substrates [19,20]. Frequently, these compounds are recognized for their ionization properties and notable water solubility, leading to their facile dissemination into both the surroundings and human physiology [21]. The intricate aromatic structures of these substances pose a challenge for biodegradation and render them inert, thereby rendering their elimination a more arduous and laborious task. In contrast to metal ions, dyes can be classified in various ways. The most prevalent classification method is based on the charge exhibited upon dissolution, which leads to the formation of three distinct groups: anionic (inclusive of reactive, acid, and direct dyes), cationic (encompassing all basic dyes), and non-ionic (comprising disperse dyes). Dyes can be categorized into acid and base based on the various associated groups that dictate the hue of the color. Acid dyes are anionic chemicals containing acid moieties in their molecular structure such as sulfonic $SO_3^{2-}$ and carboxylic $-COO^-$, while base dyes are cationic ones presenting quaternary amine groups $-NH_4^+$ [22]. Another systematic method of classification is the color index, which is related to the chemical structure of the dye substance; however, due to the complexity of nomenclature from the chemical structure, the classification based on color application is the most preferable [13]. With respect to chemical structure, a variety of groups such as azo, diazo, anthraquinone, nitro, diphenylmethane and triphenylmethane, indigoid and thionindigoid, anthraquinoid, xanthene, phthaleins and metal complex dyes are known (Figure 1). Meanwhile, the mode of application and substrate-based scale classifies them into reactive, acid, base, vat, direct, solvent, disperse, and azoic dyes [23,24].

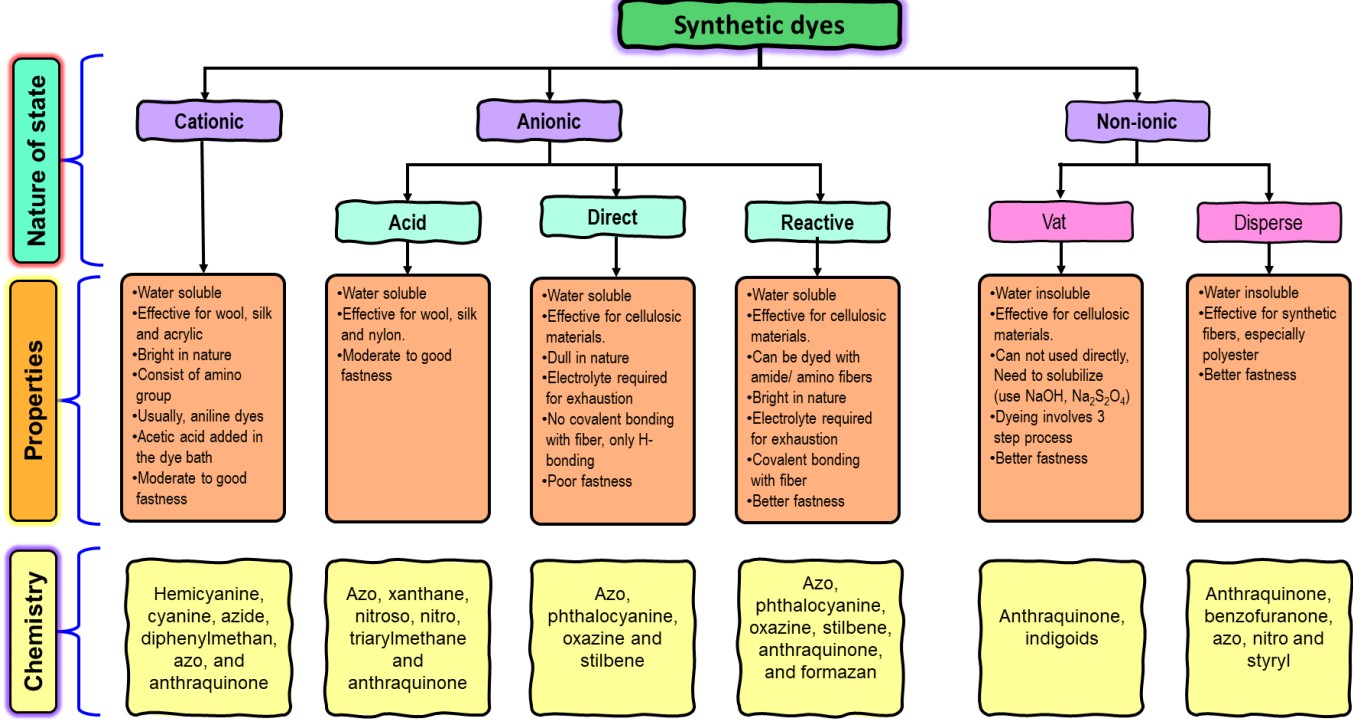

**Figure 1.** Classification of dyes used in the textile industry.

The classification of colored substances can be divided into two categories: natural and synthetic. Synthetic substances have become the predominant choice in the market due to their vast array of available colors and cost-effectiveness, as noted in [21]. The utilization of synthetic dyes, which are derived from benzene and its derivatives, has supplanted the use of conventional natural dyes, leading to the development of over 10,000 dyes

with varying chemical structures and characteristics [25]. The compounds exhibit intricate conjugated architectures that pose challenges in terms of their elimination [25]. Certain dyes, such as azo dyes, possess a high degree of toxicity and carcinogenicity because of their toxic metabolites and aromatic amine byproducts. The removal of anionic and non-ionic dyes through conventional techniques poses a challenge due to their high water solubility and resistance to degradation of non-ionized fused aromatic rings, respectively. In the interim, it has been observed that biological techniques are not entirely effective in the complete elimination of reactive and acidic dyes [7]. In general, azo dyes exhibit a high susceptibility to degradation at their azo N=N linkage, leading to the formation of hazardous aromatic byproducts during treatment. Conversely, other categories of dyes are characterized by a low degradability, which limits the range of viable treatment options. Prior to discharge into the aquatic environment, it is imperative to subject the wastewater generated by the textile sector to appropriate treatment measures. Figure 2 illustrates the relationship between the denim factories and the resultant wastewater, which significantly contributes to the contamination of the Noyal River in Tirupur, India, as well as the adjacent agricultural areas. The production of denim entails the discharge of wastewater containing various pigments (Figure 2a), subsequently leading to the pollution of nearby water bodies (Figure 2b) and ultimately culminating in the contamination of the river (Figure 2c,d). The impact of the situation on agricultural activities is evident in the images, as depicted in Figure 2e. Notably, the groundwater is significantly affected, as seen in Figure 2g,h.

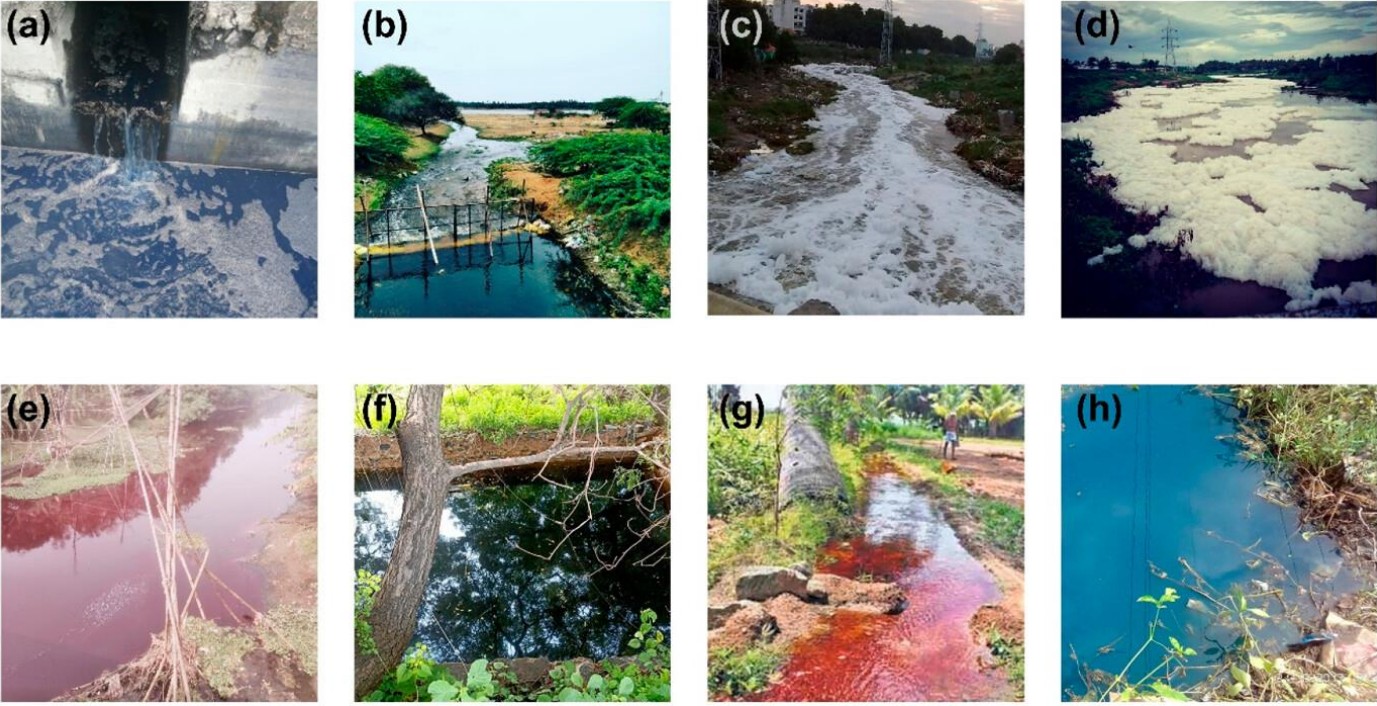

**Figure 2.** Textile effluent from industry (**a**) and in the water streams (**b**) that flow in the Noyal River (**c–e**) in India, and the influence of these effluents on the agricultural land (**f–h**) (pictures taken in April 2022 and reprinted with the permission of ACS [26] licensed under CC-BY 4.0).

In recent years, there has been a significant focus on the removal of dyes from wastewater owing to their hazardous properties and conspicuousness also now this wastewater also contains the microplastics [27,28]. The presence of dyes in wastewater can have severe consequences, including the disruption of photosynthesis and oxygen deficiency caused by the obstruction of sunlight. These dyes can accumulate in food chains, leading to aesthetic issues in aquatic and soil environments. Moreover, their carcinogenic and mutagenic properties can result in the formation of tumors and mutations. The toxic amines and biologically non-biodegradable byproducts present in dyes can also have immunological

and dermal effects. Additionally, the presence of dyes can disrupt seed germination and hinder plant growth, as well as cause chromosomal aberrations and various diseases [23,29]. To mitigate the detrimental impact of dyes and other chemical waste generated by the textile industry on both the environment and human health, it is imperative to extract the contaminated wastewater from the hazardous effluents prior to their release into water bodies. Therefore, the subsequent sections will present a concise review of the impact of health and environmental concerns linked to these effluents. There exists a more profound discussion concerning the significance of employing sustainable methodologies for the removal of effluents using various strategies. The primary objective of this review study is to consolidate and analyze the effects of wastewater containing dyes, as well as their possible implications for human health and the environment. Additionally, the dedicated section delineates a range of ecologically friendly approaches for the management and treatment of wastewater in the textile sector, together with its respective advantages and disadvantages.

## 2. Common Treatment Methods for Textile Dyes

The primary approaches utilized in the treatment of wastewater may be categorized into three distinct groups: biological, chemical, and physical. Table 1 provides a concise overview of the methodologies and presents a comprehensive analysis of their respective merits and drawbacks. One example of physical techniques is the utilization of membrane technology. On the other hand, chemical methods encompass processes such as oxidation, coagulation, and photochemical oxidation. Additionally, biological approaches include the implementation of anaerobic/aerobic sequential processes [30]. Oxidation is a chemical process that encompasses several techniques, including bleaching, chlorination, and ozonation. These techniques include the utilization of specific chemicals such as hydrogen peroxide, permanganate, chlorine, chlorine dioxide, and ozone ($O_3$), respectively [13,24,31,32].

**Table 1.** Advantages and disadvantages of the most common techniques for removal of dyes from wastewater.

| Type | Treatment Method | Advantages | Disadvantages | Ref |
|---|---|---|---|---|
| Chemical | Oxidation | Rapid and effective for both organics and inorganics<br>Can be used for both soluble and insoluble dyes<br>No need to use microorganisms | Formation of by products<br>High energy consumption and running costs | [33–35] |
| | Ozonation | No alternation in the sample volume | Short half-life and need pretreatment | [33,36–42] |
| | Chemical precipitation | Low investment and simple process | High maintenance and required to dispose the sludge | [43,44] |
| | Electro kinetic coagulation | Economic process | High sludge generation | [45] |
| | Electrochemical treatment | Moderate metal selectivity<br>Rapid breakdown | Formation of by products<br>Require high energy | [46] |
| | Advanced oxidation with Fenton reagents as catalyst | No energy input required<br>Effective for both insoluble and soluble dyes, for wide variety of wastes treatment | Sludge formation<br>Expensive process | [47–53] |
| | NaOCL | Accelerated azo bond cleavage | Toxic aromatic amine release | [53] |
| | Photochemical degradation (based on catalyst) | Effective oxidation and lab scale applicability<br>No sludge generation | Formation of by products<br>Excessive dissolved $O_2$ is required | [37] |
| | Coagulation-Flocculation/Sedimentation | Variety of coagulants-flocculants | Expensive chemicals and no recycling | [54,55] |

**Table 1.** *Cont.*

| Type | Treatment Method | Advantages | Disadvantages | Ref |
|---|---|---|---|---|
| Biological method | Single cell organisms such as bacteria, fungi, algae and yeasts | Generally, these are more economical than chemical and physical methods. For any dye industry and as a preparatory step for removal Acceptable efficiency for low concentrations and volumes Highly effective for specific dye species | Requires large land area, less flexible in operation and design and partially to totally non-degrading to dyes | [56–58] |
| | Aerobic (presence of free DO) | Facile COD removal | Longer detention times | [59–61] |
| | Anaerobic (absence of DO) | Resistance to wide variety of dyes Steam generation via the produced biogas | Longer acclimatization phase | [60,61] |
| Physical | Membrane (ultrafiltration, microfiltration, nanofiltration, reverse osmosis) | Removes all types of dyes | Inapplicable for wastewater treatment due to the large pore size | [62–65] |
| | Adsorption | For all dye industry Regeneration of adsorbent with low loss | Only soluble dyes High energy consumption | [66] |
| | Ion exchange | | For specific applications | [67,68] |
| | Irradiation | Wide range of colorants Efficient even for low volumes | High dissolved oxygen requirement Light-resistant colorants cannot be degraded | [69] |

## 3. Effluent from the Textile Industry: Human and Environmental Issues

The effluents discharged by the textile industry in their untreated state consist of a wide array of organic contaminants, including unfixed colors, acids, alkalis, and notably, very poisonous dyes [70]. The textile business employs many categories of dyes, with azo dyes being the predominant group utilized, accounting for over 60% of the industry's usage [71]. Azo dyes are characterized by their structural composition, which includes one or more azo groups. The discharge of unfixed azo dyes into wastewater is attributed to the inefficiency of textile dyeing processes, accounting for a range of 10–50% [29,72,73]. Certain textile manufacturing facilities employ wastewater treatment methods to break down the released free azo dyes in order to mitigate their impact on the environment. Conversely, there are other industries that release untreated industrial effluents straight into water sources, hence presenting significant ecotoxicological risks and causing harmful effects on organisms (see Figure 3). Farmers in various Asian nations, such as India, Bangladesh, Vietnam, and Indonesia, have historically employed the practice of irrigating their agricultural lands with untreated industrial effluents present in wastewater [74,75]. This practice has had detrimental effects on both soil quality and crop germination rates. Furthermore, the presence of toxic chemicals in these effluents has had a significant adverse impact on agricultural productivity, which in turn has had a notable influence on the gross domestic product (GDP) of these countries [76]. The introduction of azo dyes into water bodies has been seen to have detrimental effects on light penetration, hence negatively impacting the growth and productivity of algae and aquatic plants [77]. Additionally, the presence of these colors has been found to hinder the formation of dissolved oxygen (DO) in the water. Moreover, the ingestion of dyes by fish and other creatures can lead to the metabolic conversion of these substances into hazardous intermediates inside their systems, so exerting detrimental effects on the well-being of both the fish and their predators [78]. Azo dyes present in industrial effluents can potentially come into contact with humans and other mammals through two primary routes: oral consumption and direct skin contact [79]. The intestinal microflora present in the human gastrointestinal tract is responsible for the conversion of azo dyes into amino acids that possess toxic properties. These toxic amino acids have detrimental effects on numerous tissues inside the human body [70,80].

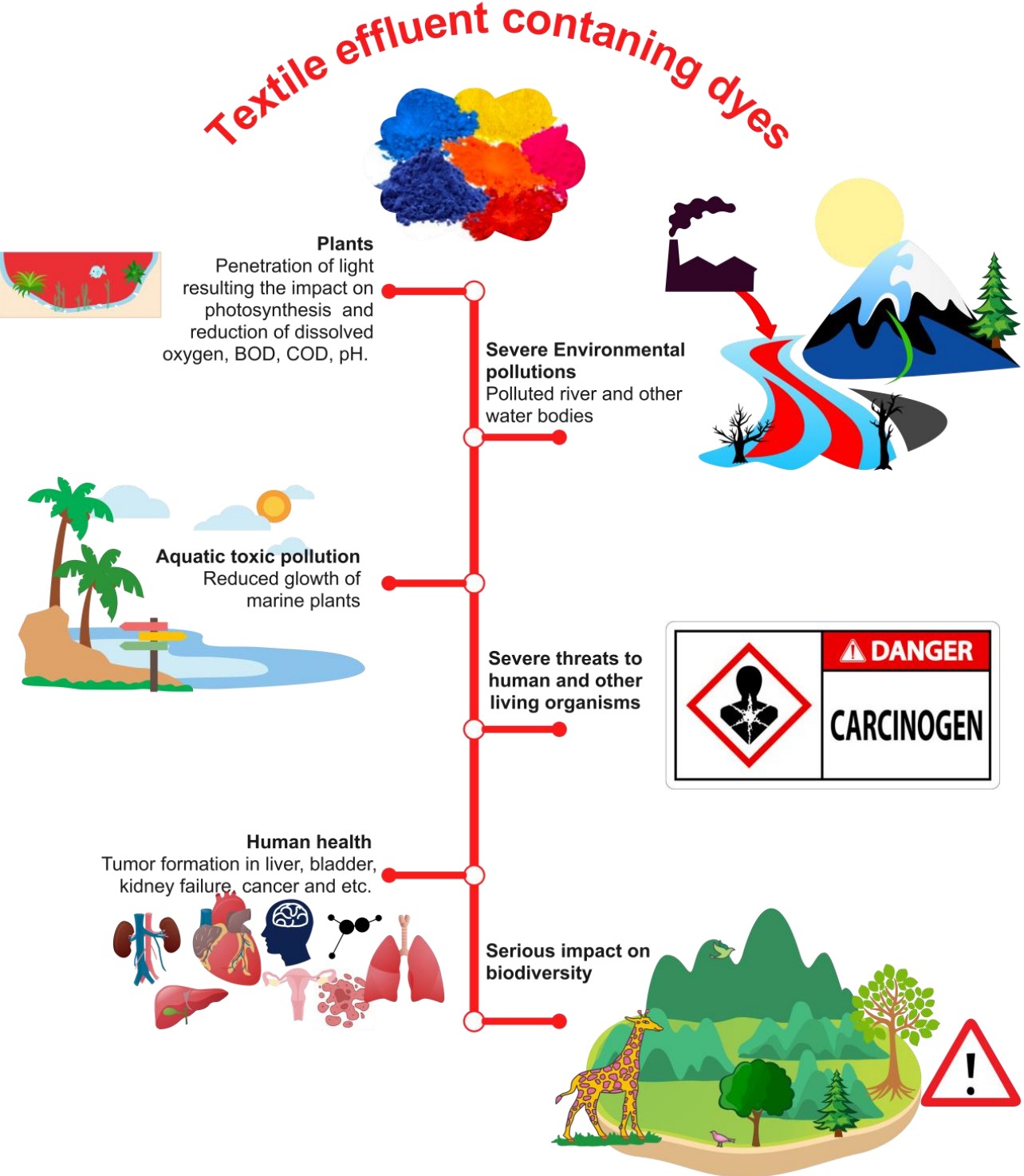

**Figure 3.** Influence of dye-containing textile wastewater on the environment and health hazardous.

*3.1. Environmental Consequences*

The textile sector disposes of significant quantities of untreated wastewater which largely contains azo dyes and several other organic contaminants (Figure 3). Nevertheless, it should be noted that azo dyes undergo degradation either before or after being disposed of, resulting in treated effluents containing amino acids that are potentially more harmful than the original chemicals [71,81–83]. Conversely, untreated wastewater has a diverse array of detrimental effects on aquatic ecosystems and creatures. The introduction of textile dyes

into aquatic habitats has been found to have adverse effects on the plants inhabiting these environments. One prominent natural concern associated with dyes is to the phenomenon of sunlight absorption and reflection in water [84,85]. It serves as a barrier that inhibits the penetration of light into the photic zone of the aquatic environment. Consequently, notable ecological ramifications ensue, including alterations in the characteristics of aquatic ecosystems and reduced photosynthetic activity relative to aquatic vegetation. Moreover, the presence of these waste liquids has been found to be associated with adverse health effects in humans, such as allergies, dermatitis, skin irritations, cancers, and mutations. Consequently, these waste liquids can contribute to the degradation of water quality, leading to the development of toxic properties, as evidenced by changes in odor and color [86,87].

As significant primary producers in freshwater and marine environments, microalgae are vital to aquatic ecosystems because they provide food for everyone from tiny zoo-planktons to gigantic whales [88,89]. Particularly in aquatic ecosystems, the single-celled tiny algae known as microalgae are essential to the upkeep of the whole food chain [89]. Nonetheless, in aquatic environments, color pollution impedes the development of microalgae and interferes with the trophic transfer of nutrients and energy. The significant quantity of textile dyes discharged into water sources affects algae development. Algae are a great indication of pollution in toxicological studies because they are more vulnerable to pollutants than other aquatic species [88–92].

### 3.2. Impact of Textile Dyes on Human Health

Fish and other aquatic animals, which are widely acknowledged as a substantial protein source for human consumption, has the potential to ingest dyes through their diet [93,94]. The wastewater generated by the textile industry has significant coloration, a fluctuating pH, and contains various salts, alkalis, and acids, which contribute to elevated levels of biological oxygen demand (BOD), chemical oxygen demand (COD), total organic carbon (TOC), and suspended solids (SS) [95]. In general, the presence of SS hinders the passage of water across the fish's gills, impeding the exchange of gases and potentially leading to reduced growth or mortality. In addition, prolonged exposure to textile effluents was found to diminish fish feed consumption, thereby leading to a decrease in the growth rate [96]. The genotoxic effects of reactive azo dyes on adult fish involve the promotion of erythrocytic micronuclei development, which is dependent on both the dose and duration of exposure. In fingerlings, the creation of gill micronuclei is also influenced by the duration of exposure to these colors. Fish are prone to a variety of illnesses due to the detrimental impact of hypoxia on their immune system and physiological responses (see Figure 4). Consequently, the presence of contaminated fish exerts a substantial influence on human well-being. Textile dyes are extremely deadly and include aromatic chemicals that have the potential to cause cancer [97–99]. They have been connected to a range of disorders in both humans and animals, including dermatitis and issues with the central nervous system [100–102]; Figure 4 lists these ailments in humans. Typically, there are two paths, which include textile dye ingestion or inhalation can irritate the skin and eyes [26,70,103], particularly if it occurs in dusty conditions [70]. Persons who work with reactive dyes run the risk of experiencing allergic responses, including occupational asthma, allergic conjunctivitis, and contact dermatitis. Textile dye genotoxicity is the biggest possible long-term risk to human health [97,98,100,104,105]. Certain dyes have the potential to cause mutagenic reactions; one such dye is Disperse Red 1 [106].

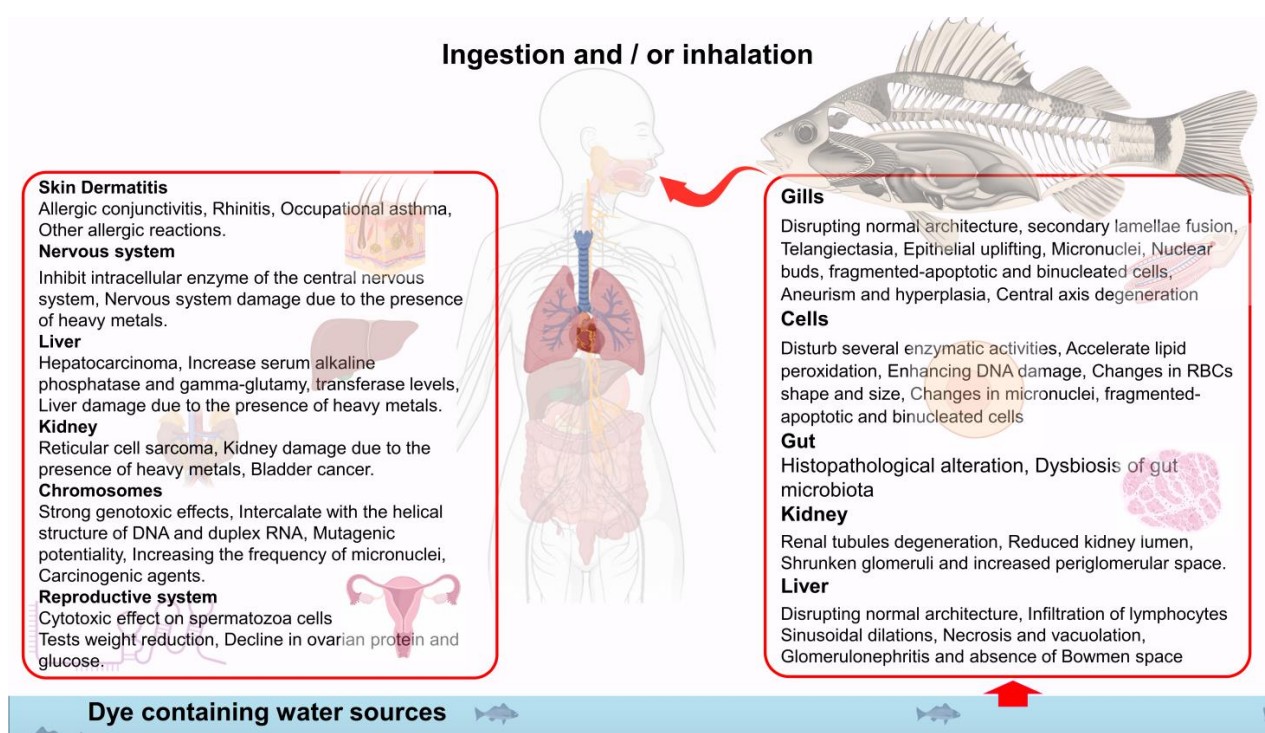

**Figure 4.** Negative impacts of textile dyes containing wastewater and their influence on aquatic life and human health.

## 4. Sustainable Wastewater Treatment for the Remediation

### 4.1. Bioadsorbents in Wastewater Treatment

The utilization of conventional chemical coagulation methods results in the generation of sludge, which is then disposed of in landfills. This practice has been found to contribute to the emission of harmful components, including gases that have the potential to contribute to global warming. Additionally, the disposal of this sludge in landfills has risks such as landfill leaching and contamination of groundwater [107]. The introduction section of this article discusses the environmental threat posed by textile effluent containing high levels of color, BOD, COD, TDS, and TSS. Biological treatment is preferred over chemical treatment for sustainable treatment. Generally, the presence of complex groups in dyes, along with the recalcitrance of organic pollutants and their low degradability, restrict the efficacy of biological treatment methods [108]. Therefore, on this occasion, bioadsorbents play a significant role in the dye and heavy metal removal. The exploitation of domestic and agricultural wastes as adsorbents has emerged as a convenient alternative. Numerous adsorbents derived from biomass wastes have been created and utilized as very effective agents for the removal of various pollutants from water and wastewater. These waste materials have been used either in their original form or following suitable modifications. Various agricultural and food waste materials, such as Azolla [109], banana peel [110–113], cabbage waste [114], chitosan [115–118], citrus peel [119,120], Citrus limonum leaves [121], corn cob [122], orange peel [123,124], peanut hull [125], rice husk [126,127], sawdust [128], and sugar cane bagasse [129] have demonstrated successful utilization as adsorbents for the purpose of eliminating diverse types of contaminants.

Adsorption generally convert the pollutants from a liquid to a solid phase. This technique has several advantages, including simple, cost-effectiveness, convenience of operation, non-toxicity, and reactive surface atoms. Bioadsorbents are frequently employed for the treatment of textile effluent water owing to their economical, eco-friendly, locally accessible, sustainable, efficient, renewable, and readily disposable characteristics. They surpass commercially available activated carbon in terms of quality, rendering the latter's high cost unjustifiable. Inexpensive sorbents possess a notable ability to absorb certain

dyes, particularly reactive dyes, leading to the accumulation of significant amounts of hydroxylates in wastewater as a result of inadequate fixation of the dyestuff. Adsorption is advantageous over alternative approaches due to its simplicity, cost-effectiveness, ease of operation, non-toxic nature, presence of reactive surface atoms, and large surface area [130]. Currently, a global revolution is underway advocating for the recycling of organic wastes from agriculture, forests, and industries into economically viable products [130]. Some of the commonly used bio adsorbents and their nature of activity in treating textile effluent water are explained below.

The peel of *Citrus limetta* has been shown to be a cost-effective adsorbent for the removal of various colors [119]. Every year, a significant proportion of citrus fruit (~40% to 60%) is discarded in landfills. Research indicates that the global citrus processing industry generates a substantial amount of trash, estimated at approximately 120 million tons [120], creating serious ecological issue. As an example, orange peels are employed for the removal of 1-naphthyl amine dye from wastewater generated by the textile industry. The findings of the study indicated that the adsorption capacity of the peel waste had a positive correlation with the concentration of dye ions. Additionally, it was observed that the percentage of dye ion removal also rose as the original dye ion concentration increased. Furthermore, the utilization of orange peels in the preparation of activated carbon has proven to be effective as an adsorbent for the removal of MB [131]. Banana fiber is an economically accessible and abundantly available material, owing to its substantial cultivation and extensive presence as a crop, with a global count over 25 billion banana or plantain trees [132]. Banana powder has demonstrated promising potential as a biosorbent for the removal of MB dye. This is attributed to the presence of many functional groups on the surface of banana particles, as well as their uneven morphology [133]. Another study found that banana peel is particularly successful in removing reactive dyes, with 90% of the dyes being removed in 5 min [134]. The utilization of ash derived from banana stem as a potential bio adsorbent for dye removal has promising results. This is evidenced by its ability to achieve a 95% removal efficiency for MB dye [135]. The effectiveness of banana stem ash may be attributed to its diverse array of components and functional groups, as well as its rough and porous surface characteristics. Recent research provides further evidence supporting the removal of 91% of color from the Banana stem [136]. Some of the resent studies confirms that the waste extraction from coffee waste shows promising adsorbents for the dyes [137].

Coconut coir dust refers to a lightweight, porous particle that is separated from the husk during the process of fiber extraction. The weight of coir dust accounts for approximately 35% of the total weight of coconut husk. Coconut coir is comprised of cellulose, lignin, pectin, and hemicellulose. The presence of hydroxyl groups in cellulose and lignin facilitates the adsorption of dyes [138]. Bio chars produced from coconut coir have enhanced dye adsorption capabilities due to their significantly higher specific surface area [139]. The research focuses on investigating the efficacy of coconut shell-activated carbon as a means of removing direct yellow DY-12 dyes. The study demonstrates that the adsorption process is particularly effective under acidic pH conditions. The findings of the study indicate that the process of adsorption exhibits heterogeneity, characterized by the formation of many layers. Furthermore, the adsorption process was seen to be endothermic in nature and occurred spontaneously [130]. Once tea has been prepared, the residual leaves are classified as waste, similar to other forms of biomass. The abundant availability of this waste has led to the increased interest in utilizing discarded tea leaves as an adsorbent [140]. Given the abundance and easy accessibility of this trash, its conversion into an adsorbent is economically viable, offering the added benefit of waste management. The utilization of raw tea waste, as well as its chemically and magnetically modified forms, in conjunction with activated carbon, has been widely employed for the remediation of water contaminated with dyes. In this study, a batch scale reactor was utilized to manufacture and apply tea powder for the purpose of removing MB from an aqueous solution. The effectiveness of adsorption was seen to improve with longer contact time, higher solution pH values, and increasing dose of waste black tea powder [141]. The residual tea waste possesses a

significant calorific value, making it suitable for utilization in steam generation within the textile sector following appropriate saturation [142].

The different form of chitosan (i.e., nanoparticles, derivatives, nanofilms, and nanofibers) is employed as a bio adsorbent. This application aims to substitute activated carbon in the pre-treatment of textile effluent, with a specific focus on the removal of metal ions, particularly chromium, as well as colors (Figure 5a). The ability to repeatedly utilize these bio adsorbents with diluted NaOH while maintaining the same level of efficacy is noted, rendering it an intriguing aspect [143]. Cactus juice and aloe vera juice were employed as flocculants for the treatment of textile effluent [144]. The color removal efficiency achieved above 85%. Furthermore, the removal efficiencies for total solids, suspended particles, and dissolved solids were found to be 90% [145]. The efficacy of water chestnut peel in the removal of cationic RhB shows promising results [146] Table 2 provides a comprehensive overview of the prior studies conducted on the treatment of textile wastewater using bioadsorbents. In recent studies, researchers have shown that chitosan modified cellulosic non-woven fabric has superior color removal capabilities for Reactive Red 198 [147]. Additionally, these non-woven fabrics have shown great potential as materials for dye removal in the future [148,149].

Various plant-based waste materials and biomasses have been found to have significant efficacy in the adsorption and retention of dyes (Figure 5b). The primary constituents of plant leaves encompass cellulose, hemicellulose, pectins, and lignin, and additionally it contains many functional groups, such as carboxyl, hydroxyl, carbonyl, amino, and nitro, which can interact with the functional groups of the dyes [150]. The adsorption of Acid Orange 52 (AO-52) dye using *Paulownia tomentosa* Steud leaves biomass showed promising results [151]. In a separate investigation, the adsorption of Acid Red 27 (AR-27), an anionic dye, was examined utilizing hyacinth leaves [152]. Basic Red 46 (BR-46) dye exhibited strong affinity towards pine tree leaf-based adsorbents [153]. Ashoka leaf powder exhibited interactive behavior towards rhodamine B (RhB), Malachite Green, and Brilliant Green dyes [154]. A novel lignocellulosic biosorbent material, obtained from fully developed leaves of the sour cherry plant (*Prunus cerasus* L.), has remarkable efficacy in the removal of Methylene Blue and crystal violet dyes [155]. The coffee waste demonstrates a characteristic three-dimensional carbon structure, with a rough surface and a porous system that allows it to function as a promising adsorbent for the removal of anionic CR and RB5 dyes from aqueous solutions [156]. The experimental findings indicate that the utilization of powdered lemon leaves resulted in the removal of Malachite Green up to a maximum efficiency of 82.21%. The highest sorption capacity ($q_{max}$) of lemon leaf powders is 8.08 mg/g [157]. In another study, the cationic amino modified banana leaves show the excellent sorption for Congo Red (CR) dyes [158]. Table 2 presents a comprehensive overview of recently studied bio adsorbents, including their respective adsorbent capacities in relation to various dyes.

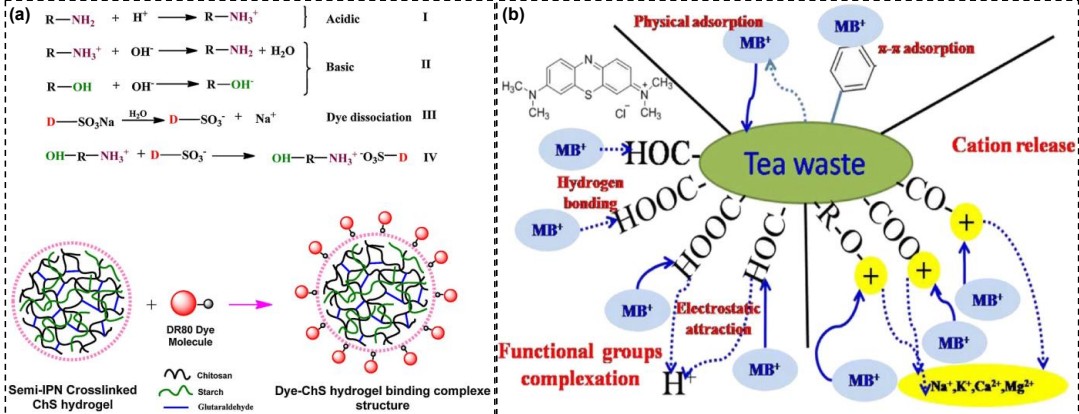

**Figure 5.** Different types of bio adsorbents for the dye removal; chitosan (**a**) (reprinted from [159]); tea waste (**b**) (reproduced from [160] as distributed by Creative Commons Attribution License).

**Table 2.** Studies on bio adsorbents on textile wastewater treatment.

| Name of Adsorbents | Performed Dyes | Adsorption Conditions | Removal (mg/g) | Refs. |
|---|---|---|---|---|
| Potato peel-based sorbent | Direct Blue 71 | pH 3 | 1704 | [161] |
| Rice husk ash | Brilliant Green dye | pH 4–10 | 66 | [162] |
| Sunflower stalk | Basic Red 9 dye | - | 317 | [163] |
| Cane pith | Basic Red 22 dye | pH 4.1 | 941.7 | [164] |
| Bagasse | Basic Red 22 dye | pH 4 | 942 | [165] |
| Enosis siliqua shell powder | - | - | 797 | [166] |
| Glutaraldehyde cross-linked magnetic chitosan beads | Direct Red 23 | pH 4 | 1250 | [167] |
| Popcorn derived activated carbon | Methyl Orange | pH 2–11 | 2090 | [168] |
| Carboxymethyl cellulose-g-poly(2-(dimethylamino) ethyl methacrylate) hydrogel | Methyl Orange | pH 2 | 1825 | [169] |
| Non-cross-linked and cross-linked chitosan fibers | Acid Orange 7 | pH 7 | 4523 | [170] |
| Chitosan grafted with diethylenetriamine | Acid Orange 7 | - | 2108 | [171] |
| Chitosan grafted with poly(methyl methacrylate) | Reactive Blue 19 | pH 3 | 1498 | [172,173] |
| Chitin nanofiber-/nanowhisker-based hydrogels | Reactive Blue 19 | pH 1 | 1331 | [172] |
| Cationic cellulose nanocrystals-chitosan film (nanocomposite) | Reactive Blue 19 | pH 3 | 1320 | [174] |
| Hollow zein nanoparticles | Reactive Blue 19 | pH 9 | 1016 | [175] |
| Chitosan films | Reactive Blue 19 | pH 6.8 | 822.4 | [176] |
| Template ECH cross-linked chitosan nanoparticles | Reactive Black 5 | pH 3 | 2941 | [177–179] |
| Chitosan beads cross-linked with epichlorohydrin | Reactive Black 5 | pH 3 | 2043 | [180,181] |
| Glutaraldehyde cross-linked chitosan beads/microparticles | Reactive Black 5 | pH 10 | 1927 | [182] |
| Chitosan cross-linked with sodium edetate | Reactive Black 5 | pH 3 | 1648 | [183] |
| Chitosan hydrogel | Reactive Black 5 | - | 1560 | [184,185] |
| Mango bark powder | Malachite Green dye | pH > 6 | $4.22 \times 10^3$ mol/g | [186] |
| Calcium-rich biochar | Malachite Green dye | Neutral and alkaline pH | 12,502 | [187] |
| Pigments-extracted macro algae derived biochar | Methylene Blue | - | 5306.2 | [188] |
| Azolla-derived hierarchical nanoporous carbons | Methylene Blue | - | 4448 | [189] |
| Banana | Reactive Blue 235 Methyl Red, Malachite Green | - | - | [190] |
| Activated surface of banana and orange peels | Reactive Red 24 | - | - | [191] |
| Waste tea residue | Acid Blue 25 | - | - | [142] |
| Palladium nanoparticles synthesized from peel waste of cotton boll | Toxic azo dye | - | - | [192] |
| Wheat husk waste | Textile effluent water | - | - | [193] |

### 4.2. Dye Removal by Biological Methods

Although it is true that certain microorganisms can degrade auxochromes and chromophores found in dyes, hence facilitating the removal of organic materials from textile waste, it is worth noting that some of these microorganisms are also capable of mineralizing colors into carbon dioxide and water (see Figure 6). The rationality of color removal in biological processes, even conventional ones, has not been empirically shown Figure 6. The rate of removal is contingent upon several factors, including the concentration of $O_2$, the ratio of organic load to microorganism load and dye load, and the temperature range [58,194]. Table S1 presents a comprehensive compilation of the merits and demerits associated with diverse biological techniques employed in the elimination of dyes.

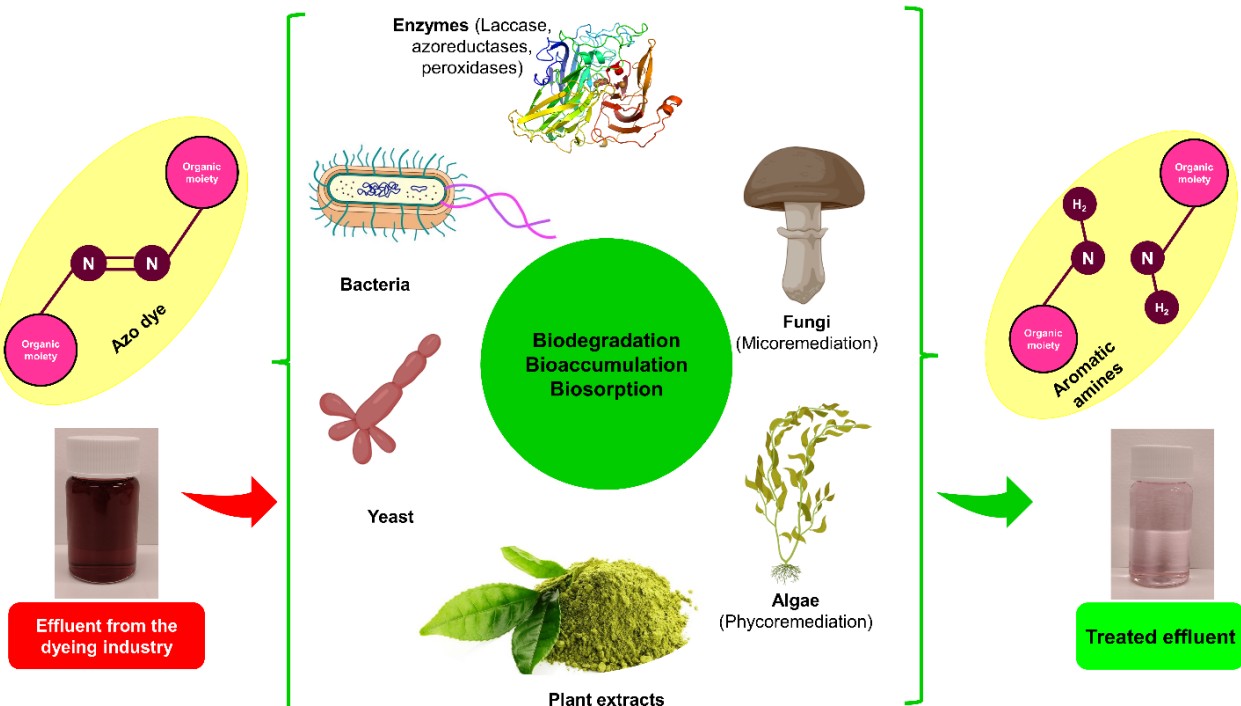

**Figure 6.** Treatment of textile wastewater by biological and biosorption methods.

4.2.1. Biological Route of Dye Decolorization

Enzymes are employed in the field of biological remediation because to their multifunctionality, effectiveness, and capacity to break down organic waste materials. The effectiveness of oxidative enzymes, such as laccases, peroxidases, and tyrosinases, has been demonstrated in the conversion of hazardous waste into insoluble compounds that may be easily separated [12,57,58,195]. The utilization of microbial enzymes for the degradation of azo dyes has been extensively investigated. Laccases are a class of enzymes that possess copper ions within their active sites, enabling them to facilitate the oxidation reactions of phenolic compounds and aromatic amines. Peroxidases are a class of enzymes that include heme and facilitate the oxidation of organic molecules using hydrogen peroxide as a co-substrate. In contrast, azoreductases are a class of enzymes capable of reducing azo dyes to their respective aromatic amines, resulting in reduced toxicity compared to the original molecules [196,197]. Laccases have demonstrated significant promise in facilitating the oxidation of a wide range of substituted phenolic and non-phenolic chemicals [198]. These organisms are found in several ecosystems, encompassing fungi, plants, and bacteria, exhibiting a broad distribution. Laccases are enzymes that exhibit a notable characteristic of not necessitating peroxidases for their catalytic activity. Instead, they employ molecular oxygen as the primary electron acceptor, rendering them highly prevalent in the enzymatic breakdown of azo dyes. Nevertheless, the economic viability of enzymes has been hindered by their inherent instability, variable activity, and labile characteristics. Enzymatic treatment of effluent water has the potential to achieve a significant reduction in coloration, with removal rates of up to 90% [199].

Bacteria can be employed to remove dye decolorization in specific cases, resulting in a 30% reduction in effluent toxicity [200,201]. Bacteria and fungi exhibit short lifespans and instability, while enzymes, although more effective, are associated with higher costs. However, stability concerns persist in both bacteria and fungi as well [83,202]. In the context of bacterial color biosorption, it has been shown that corynebacterium glutamicum exhibits potential as a biosorbent for Reactive Black-5, with a sorption capacity of 257 mg/g at a pH4 [203]. There exist two prevalent categories of microscopic organisms, namely Gram positive and Gram negative. Microscopic organisms participate in the formation of a

dense peptidoglycan layer that is interconnected by amino-acidic linkages. Polyalcohols, which are lipids linked to form lipoteichoic acids known as Gram positive, are present within the cell divider [204]. The degradation of dyes by different bacterial biomass is comprehensively elucidated in the provided Table S2.

Numerous research groups have investigated the ability of microorganisms to digest azo dyes. Pseudomonas bacteria do not readily use azo dyes in aerobic environments. Despite the interruption of metabolic pathways by the intermediates generated during these degradative steps, the trash was not subjected to mineralization. In environments without oxygen, a multitude of microorganisms are known to enzymatically degrade azo dyes through the activity of soluble, unspecified cytoplasmic reductase enzymes, commonly referred to as azoreductases. Enzymes facilitate the production of colorless aromatic amines, which have the potential to exhibit mutagenic, fatal, and potentially carcinogenic effects on organisms. The existing body of research suggests that there are many supplementary methods that may be employed for the reduction of azo dyes. A wide range of microorganisms have the ability to biodegrade both sulfonated and non-sulfonated azo dyes under anaerobic conditions. Furthermore, a multitude of highly charged atomic, polymeric, and sulfonated azo dyes have an inability to traverse the cell membrane. Therefore, the capacity to remove dye is not attributed to the intracellular accessibility of the azo dye [204].

### 4.2.2. Fungi

Fungal mycelia possess an advantage over unicellular organisms due to their ability to solubilize insoluble substrates, hence producing extracellular enzyme catalysts. Organisms have enhanced enzymatic and physical interactions with their environment because of an increased ratio of cell surface area to volume [205,206]. Several types of fungi, including white-rot fungi, *Aspergillus niger* [206], *Rhizopus arrhizus* [207], and *Rhizopus oryzae* [208], have been found to possess the ability to degrade a wide range of colors.

### 4.2.3. Algae

Algae possess an abundance of enzymes and other compounds that contribute to the process of dye decolorization in textile effluent. Some algae can metabolize dyes through enzymatic processes, leading to the breakdown and detoxification of these substances. *Chlorella vulgaris* has been effectively used to remove dyes, including Congo Red, Brilliant Blue R, and Remazol Brilliant Blue R, from wastewater. Algae have unique metabolic abilities that enable them to efficiently remove or break down contaminants, such as dyes [209,210]. *Cosmorium* sp. has been investigated as a biosorbent for the removal of Malachite green (MG) dye, resulting in a significant removal efficiency of 92% [211]. Several studies have demonstrated the considerable influence of algae on the process of decolorization. For instance, *Cosmarium* sp. achieved a noteworthy 74% elimination of malachite green [212]. Similarly, *Azolla rong pong* exhibited decolorization rates of 30% for Acid Green-3 [213] and 43% for Acid Blue-15 [214]. In a separate investigation, *Ulva prolifera* demonstrated a remarkable 96% efficacy in the removal of Acid Red-274 colorant [215].

### 4.2.4. Enzymes

Enzymes are frequently employed for effective dye removal from wastewater, with fungi being the primary source of these enzymes. Additionally, laccase is an enzymatic protein that belongs to the class of copper-containing polyphenol oxidases, which are synthesized by many species of bacteria and fungi. The utilization of this technology has been employed for the decolorization of azo dyes. Peroxidase is an enzymatic catalyst responsible for the decomposition of hydrogen peroxide ($H_2O_2$) and the promotion of oxidation reactions involving various substrates through the use of molecular oxygen. The utilization of lignin peroxidase has been previously applied for the purpose of removing CR dye. Furthermore, it has been utilized for the purpose of decolorizing Reactive Orange 16 [216–219].

The mutant laccase enzyme was elevated in *Escherichia coli*, resulting in the decolorization of indigo carmine by more than 92% (see Figure 7a) [220]. Azoreductases are a distinct class of flavoenzymes that have the potential to facilitate the reduction of azo bonds (–N=N–) present in the aromatic dyes by hydrolysis of azo bonds in the presence of oxygen or in the absence of oxygen, hence participating in the metabolic breakdown of dyes (see Figure 7b) [221]. This process yields aromatic amines, which are subsequently eliminated by microbial enzymes including mono- and di-oxygenases, as well as hydrolases [222]. In addition, azo dyes are capable of reducing the toxicity of nitro-aromatic compounds by reducing their concentration. Azoreductases are typically pH-stable within the range of 5–9, and their activity is at its peak under physiological conditions [197]. In 2019, Dong et al. [223] determined that the use of azoreductases derived from Streptomyces species shown efficacy in the elimination of MR from wastewater. In the same year, Sherifah et al. [224] employed the utilization of *Kluyveromyces dobzhanskii* bacterial laccase for the purpose of enzymatically degrading MG and MR dyes. In a separate investigation, yeast laccase derived from *Yarrowia lipolytica* was utilized to enzymatically degrade Bromocresol Purple, Safranin, Bromothymol Blue, and Phenol Red. Additionally, the process of isolating laccase from the edible fungi species *Agaricus bisporus* was conducted, with the intention of using this enzyme for the purpose of enzymatically degrading Acid Blue [225].

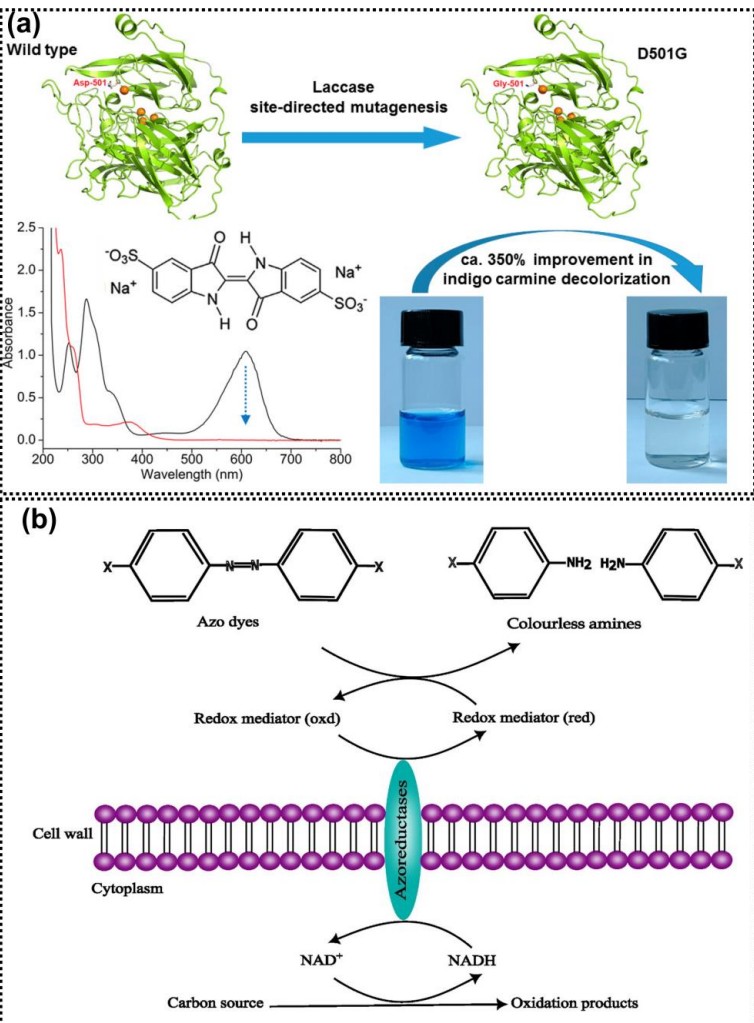

**Figure 7.** Site-directed mutagenesis of laccase from *B. amyloliquefaciens* for improving indigo carmine decolorization (**a**) (reproduced from [220] as distributed by Creative Commons Attribution License). Azoreductase on azo dyes with the formation of colorless amines (**b**) (reprinted from [221]).

### 4.2.5. Bacteria

Bacterial isolates provide a viable and ecologically sustainable approach for the breakdown of dyes. Bacterial isolates possess enzymatic systems that enable them to eliminate dyes via degradation or biosorption mechanisms [226]. Bacterial organisms have the advantageous characteristic of exhibiting shorter development durations, in addition to their capability to breakdown and mineralize dyes [227]. Numerous investigations have employed bacterial strains for the purpose of degrading and decolorizing diverse textile effluents that contain azo dyes, such as MO (i.e., color removal of 95%), Reactive Yellow (84%), and Reactive Black 5 (100%) [228–233]. Additionally, it shows better color removal with anthraquinone-based dyes like Remazol Brilliant Blue R [234], disperse blue 2BLN (decolorization rate of 93.3%) [235].

### 4.3. Membrane Separation

Membrane separation technology is commonly employed for the treatment of effluents generated by textile dyeing processes. During the filtering process, the micropores included in the membrane filter effectively separate the organic compounds from the effluent by utilizing selective membrane permeability. The classification of this phenomenon encompasses four distinct categories, namely ultrafiltration (UF), nanofiltration (NF), reverse osmosis (RO), and forward osmosis. The process of separation may be effectively achieved by the utilization of UF, which has shown great potential as a technique. The elimination of dissolved compounds occurs at a reduced transmembrane pressure through the utilization of UF. The utilization of polyelectrolyte complexes, in conjunction with cellulose acetate and inert polymers, is applied in the production of UF membranes that exhibit the capacity to efficiently regulate flow. The normal range for pore size is between 0.001 and 0.02 μm. NF is an intermediate technique between reverse osmosis and ultrafiltration, characterized by the use of membranes with nanometer-scale pores (0.5–10 nm) and operating at pressures of 5–40 bar. NF is a very sophisticated membrane-based technique that demonstrates remarkable efficacy in the removal of heavy metals [236–238]. The membranes of NF possess a thin outer layer that is typically non-porous, operating at the nanoscale, and exhibiting a high level of permeability [62]. One of the primary benefits of NF is its reduced energy consumption, which leads to a higher efficiency in the removal of contaminants [239]. Presently, several textile industries employ RO as a means of treating their effluent. RO is categorized as a membrane-based technique. The RO membranes effectively capture suspended particles through their small pores, hence mitigating fouling. The pre-treatment procedure plays a crucial role in the regulation of turbidity levels and fouling tendencies [62]. Figure 8 illustrates the classification of membrane filtering techniques together with their respective advantages and disadvantages.

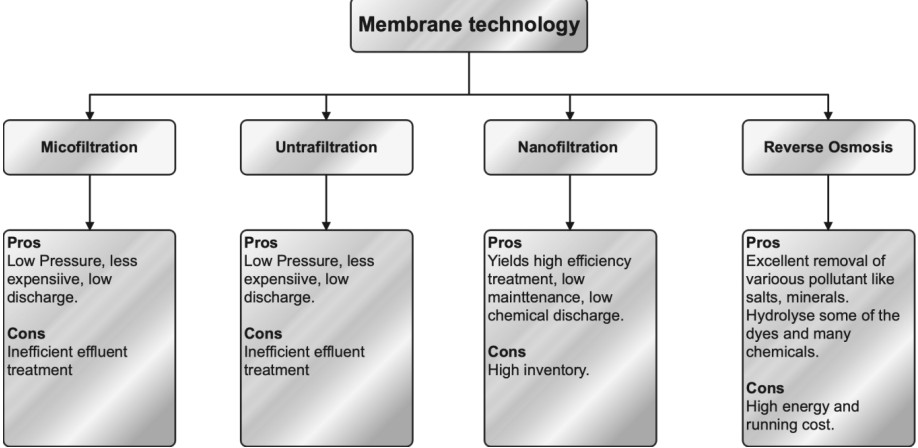

**Figure 8.** Classification, advantages, and disadvantages of membrane technology.

Figure 9 depicts the design of a textile wastewater treatment plant, presenting a comparative analysis between the conventional approach and the membrane bioreactor-nanofiltration (MBR-NF) technique. The use of a plant-based system that incorporates membrane bioreactor (MBR) and NF technology has promise in reducing the dependence on anaerobic, aerobic, coagulation, and decolorization dosing/sedimentation tanks [240]. The introduction and utilization of an MBR-NF treatment facility has resulted in a substantial decrease in operational costs. Moreover, this sophisticated technology has the benefit of occupying a smaller physical footprint when compared to conventional treatment facilities. Figure S1 illustrates the application of combined coagulation–flotation with forward osmosis technology for wastewater treatment. The removal efficiency in this method is characterized by its high-water flow and high recovery rate. The extent of membrane fouling is rather little; yet it engenders adverse consequences on the environment. The experimental setup employed in the forward osmosis system involved the use of a fabricated forward osmosis membrane, namely a plate-and-frame configuration, with a surface area of 10 cm$^2$ [241]. A spacer-free rectangular canal is installed on both sides of the membranes. In the first stages of forward osmosis, the amount of wastewater is reduced by employing osmosis to extract water from the wastewater, hence increasing the concentration of dye in the remaining solution [242].

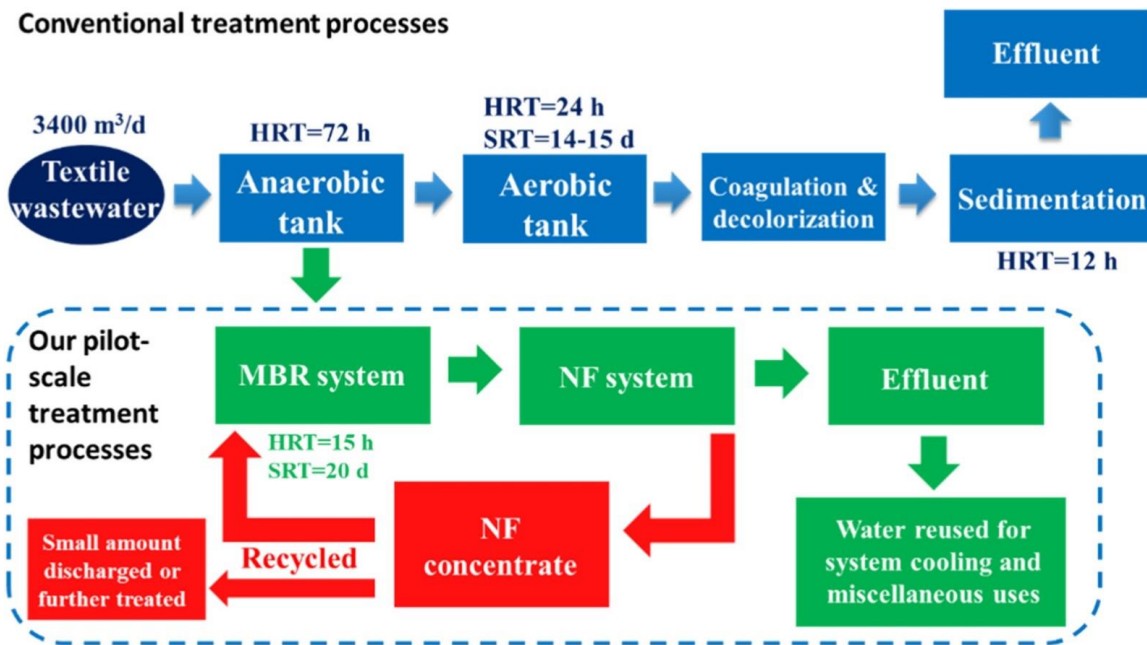

**Figure 9.** Process flow of conventional treatment and combination of MBR-NF process (reprinted from [240] with the kind permission of Elsevier).

### 4.3.1. Ion Exchange

The term "ion exchange" refers to the reversible process of exchanging ions between a liquid and a solid, without causing any significant alteration to the solid's structure. Ion exchange is a widely employed method for the widespread removal of inorganic salts and organic anionic constituents, such as antibiotics, amino acids, organic acids, and small compounds [67]. The classification of ion exchange resins is based on the functional groups present, resulting in three main types: anion exchange resin, cation exchange resin, and chelating exchange resin. The substance in question may be categorized as either natural or synthetic, and it possesses several notable advantages, including affordability, minimal equipment requirements, straightforward operation, and the absence of solvents [67]. One significant limitation of this process is the extended duration required for production, as well as the substandard quality of the resulting product. Additionally, the high pH levels and the potential transfer of dirt and contaminants from the effluent to the sludge pose

further challenges. Furthermore, the introduction of chemicals for sludge regeneration exacerbates these issues [67,68,243].

### 4.3.2. Evaporation

During the evaporation process, the concentrated textile effluent is subjected to various evaporator systems and steam within the evaporator. The attainment of the desired salt concentration or specific density is achieved only by the recirculation of the liquid during the evaporation process. The condenser is responsible for gathering the vapor and steam, and the evaporator temperature is subject to variation based on the length of the tubes. Textile effluent is commonly subjected to evaporation through two primary methods, namely solar evaporation and mechanical evaporation [195,236].

### *4.4. Other Techniques*

### 4.4.1. Granular Activated Carbon (GAC)

Carbon is a non-metallic element that is abundantly present in nature and finds extensive use across many applications in daily human existence. Graphite has a wide range of applications, including as a source of fuel, lubrication, material for pencils, electrodes, and as a means of water filtration [244]. Activated carbon refers to a kind of carbon that has been specifically engineered to possess small, low-volume holes and a significantly increased surface area. This enhanced surface area facilitates the process of adsorption or chemical reactions, hence enabling the purification of both liquids and gases. GAC refers to a specific form of carbon that is capable of being retained in a 50-mesh sieve [245–247]. This type of carbon can be obtained from various sources by different extraction procedures and with varying degrees of activation. The substance is offered in many forms, including granules, powder, and pellets. Activated carbon is commonly derived from many sources such as coconut shell, hard and soft wood, peat, olive pits, lignite, and bituminous coal using chemical or steam-based processes. The activated carbon has a surface area of 500 m$^2$/g, indicating its porous characteristics. Various studies have been conducted utilizing a range of biomass materials such as bagasse, coal, rice husk, coconut husk, nutshell, lemongrass, sawdust, cocoa shells, grape peels, and cassava peels. These biomass materials have been subjected to activation processes involving ZnCl$_2$, phosphoric acid, microwave assistance, microwave assistance combined with KOH activation, and steam pressure. The objective of these studies is to investigate the efficacy of these activated biomass materials in the removal of dye from effluent water [245–248].

### 4.4.2. The Advanced Oxidation Process (AOP)

The AOP is mostly observed in the field of water purification, but more recently, it has been employed for the remediation of textile effluents. Hydroxyl or sulphate radicals are liberated in sufficient amounts to facilitate the elimination of both organic and inorganic substances, pollutants, and to enhance the water's biodegradability. In comparison to chlorine and ozone, these substances exhibit superior performance in terms of water decontamination and disinfection. Various categories utilize the hydroxyl radical. Various methods have been employed in the field of environmental remediation, including UV-based processes, ozone treatment, Fenton reactions, and the utilization of sulphate radicals, among others, additionally UV has advantages for disinfection properties. The different advanced oxidation processes are illustrated in Figure 10a. The AOP is well recognized as a prominent technique for the treatment of industrial wastewater, owing to the considerable oxidative potential shown by ozone and the resulting generation of hydroxyl radicals (OH) [249]. Extensive research has been conducted on the application of ozone-based AOPs in both simulated and actual environmental circumstances. The use of auxiliary agents in the dye and their impact on dye degradation, as well as the influence of different salts on the process of ozonation, were investigated through the application of the AOP [250]. The AOP has gained significant popularity in the field of leachate treatment and water reuse [251]. There exist several forms of AOPs, including ozone, ozone/hydrogen peroxide, ozone/UV,

UV/TiO$_2$, UV/hydrogen peroxide, Fenton reactions, Photo-Fenton reactions, ultrasonic irradiation, heat/persulfate, UV/persulfate, Fe(II)/persulfate, and OH-/persulfate [252].

### 4.4.3. Color Removal by Fenton Oxidation

The Fenton oxidation method is a very promising technique for the treatment of textile wastewater due to its cost-effectiveness and ease of implementation [48]. The major objective of Fenton oxidation is the decolorization of the effluent, although it also possesses the ability to degrade organic pollutants (Figure 10b). Hydrogen peroxide can be employed as an oxidizing agent, either in the presence or absence of a catalyst. Notable catalysts that can be utilized include ferrous salts, Al$^{3+}$, and Cu$^{2+}$ [49]. The efficacy of Fenton's reagent has been demonstrated in the treatment of many types of industrial effluent as well as a wide range of dyes. The Fenton process demonstrates a high level of effectiveness in removing color, with an efficiency of 98% achieved at a pH of 3. Similarly, the Fenton process exhibits a significant capability for removing COD, with an efficiency of 85% achieved at a pH of 3 [49,51]. The most efficient decolorization of effluent for all dyestuffs occurs at a pH value of 3, within the range of 2.5–4. The utilization of this reduced value is attributed to the substantial production of OH [51]. When H$_2$O$_2$ and (Fe$^{2+}$) are combined under these specific pH conditions, hydroxide ions (OH$^-$) are generated by a complicated series of interconnected reactions [51,253]. Figure 10c illustrated the decolorization of an indigo-dyed pollutant to colorless.

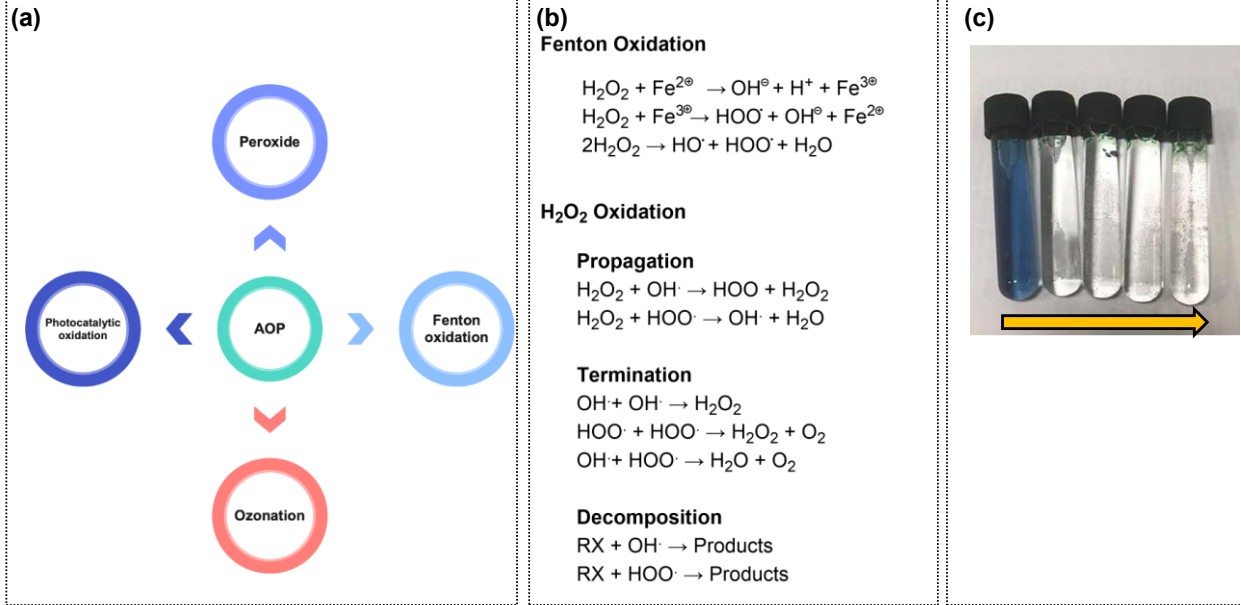

**Figure 10.** Different advanced oxidation processes (**a**) reaction during Fenton and hydrogen peroxide-based oxidation (**b**) and the color removal with different time spans of the reactions with Fenton oxidation (**c**) (reprinted [254] with the permission of Springer Publications).

### 4.4.4. Color Removal by Peroxide (H$_2$O$_2$)

Hydrogen peroxide has a high degree of efficiency and contains the OH$^-$ radical, which is accountable for both the chemical breakdown and mineralization of organic molecules, and is generated by the reaction involving another oxidant, H$_2$O$_2$. Furthermore, treatment of halogenated substances results in the generation of non-hazardous halide ions and non-toxic molecules, including carbon dioxide (CO$_2$) and H$_2$O [255]. A notable observation is that the efficiency of H$_2$O$_2$ addition in a recirculated photoreactor is significantly higher when performed in a single-step manner, as opposed to multiple-step addition [255]. Due to its short lifespan, the generation of OH$^-$ occurs in situ by the reaction induced by UV irradiation, as follows,

$$H_2O_2 + UV = 2\,OH^- \tag{1}$$

The breakdown of organic pollutants is facilitated by the $OH^-$ radical through four primary routes, radical addition, hydrogen abstraction, electron transfer, and radical combination [256,257]. The application of $H_2O_2$-UV results in the degradation of the chromophore configuration of the dye, leading to its decomposition in normal environmental circumstances. This process generates $O_2$, which may be effectively utilized for aerobic treatment [256]. The effectiveness of wastewater decolorization is enhanced in an acidic environment [256,257]. $H_2O_2$ is utilized in the oxidation of alkali, resulting in the formation of $O_2$ and $H_2O$. This process generates accessible hydrogen peroxide for the hydroxyl radical ($OH^-$). The reduction in radical production leads to a decrease in decolorization efficiency [256].

4.4.5. Ozonation

Ozonation is considered as an environmentally sustainable method for treating wastewater owing to its lack of residue production and absence of chlorinated byproducts, which are known to be toxic. This process effectively oxidizes color, odor, and bacteria without generating any detrimental substances [258–260]. The decomposition of organic compounds, detergents, and phenols into smaller molecular components is aided by the process of oxidation, which may be achieved using commercially available sodium hypochlorite. As a potential alternative, the implementation of ozonation might be regarded as a feasible solution to supplant the utilization of hypochlorite [40,261–264]. Typically, the process is conducted at alkaline conditions, characterized by a pH greater than 9, as the degradation of ozone in water is enhanced under such circumstances. The process of oxidizing inorganic compounds and dissolved organic molecules with ozone involves two distinct processes. The direct reaction of ozone molecules exhibits a higher degree of selectivity, characterized by a relatively slow reaction rate. This reaction is particularly advantageous in acidic conditions. The indirect response exhibited by free radicals, including $OH^-$ and HOO, is characterized by reduced selectivity and a preference for basic conditions. Another noteworthy characteristic is that the reactivity of the dye is enhanced when it possesses an electron-donating group at its ortho and para locations, as opposed to an electron-withdrawing group [71,265,266]. The ozonation process is impeded by the presence of salts, such as NaCl or $Na_2SO_4$. However, it is worth noting that the presence of NaCl is more undesirable compared to $Na_2SO_4$. This is since $Na_2SO_4$ generates sulfate or peroxysulfate radicals, which might somewhat facilitate the ozonation process [259]. The use of an ozonation membrane biological reactor offers a means to enhance the removal of harmful substances such as pesticides, while simultaneously reducing the reliance on traditional coagulation methods. This innovative approach also lowers the need for additional biological treatment, resulting in a simplified operational procedure [41,267]. In many instances, ozone is employed in conjunction with UV or hydrogen peroxide to achieve enhanced efficacy. The utilization of ozone in conjunction with UV radiation facilitates the activation of ozone molecules, hence facilitating the creation of hydroxyl radicals ($OH^-$). UV radiation assists in expediting the oxidation process by aiding in the completion of the process. The exclusive use of $O_3$ may result in incomplete conversion of organic compounds into $CO_2$ and $H_2O$ in some cases. In addition, the intermediate process involving $H_2O_2$ undergoes photolysis, resulting in the formation of a hydroxyl radical which subsequently decomposes the dye molecules. When $H_2O_2$ is employed in conjunction with ozone for oxidation applications, it functions as a catalyst to enhance the generation of hydroxyl radicals ($OH^-$) through the breakdown of ozone. The interaction between hydrogen peroxide $H_2O_2$ and $O_3$ has a sluggish rate under acidic pH conditions, but it undergoes a significant acceleration the in-reaction rate at elevated pH levels.

### 4.4.6. Photocatalytic Oxidation

The photocatalytic process is often regarded as the predominant method for treating textile effluent water due to its distinct advantage of optical absorption, a characteristic not shared by other AOPs. Photocatalysts could be stimulated by radiation, resulting in the generation of exceptionally reactive photo-induced charge carriers (free radicals, notably hydroxyl radicals (OH$^{\cdot}$) that engage in chemical reactions with contaminants [90,91,268]. These free radicals facilitate the oxidation of organic molecules, leading to their full conversion into non-toxic chemicals, such as carbon dioxide $CO_2$ and $H_2O$ by the absorption of photons. The mitigation of pollutants carried out under ambient temperature and pressure conditions has the potential to successfully address the issue of excessive energy consumption associated with conventional approaches, such as electrochemical technology. The systematic exploration of photocatalysis research started in the early 1970s. The Honda–Fujishima effect was investigated by Fujishima et al. [269], who employed solar energy and titanium dioxide as photocatalysts to carry out water breakdown and hydrogen reduction activities. The utilization of light as an essential factor in this process allows for the possibility of designing the reactor or photocatalyst in a manner that enables the utilization of sunlight as a cost-effective energy source to drive the reaction [270]. The photocatalysts utilized in this study must possess characteristics that are conducive to their practical application. These characteristics include ready accessibility, reproducibility, photoactivity, non-toxicity, non-corrosiveness, biological or chemical inertness, affordability, and compatibility with near UV or visible light wavelengths.

### 4.4.7. The Sequencing Batch Reactor (SBR)

The implementation of the Sequencing Batch Reactor (SBR) was undertaken with the objective of mitigating the presence of nitrogen and phosphorus in piggery waste, as well as facilitating the biodegradation of sulfonated azo and diazo reactive colors found in textile effluent [271–273]. The study has presented findings on the effectiveness of SBR in the elimination of azo dye. It has been proposed that the inclusion of aerobic bacteria capable of degrading amines in the SBR system might enhance the complete mineralization of reactive azo dye under anaerobic conditions [272,274]. According to reports, it was observed that the decolorization reached a maximum of 97%, while the elimination of COD reached up to 98%. It was also suggested that the rate at which dyes are removed by SBR is influenced by the volumetric dye loading rate [275]. At present, bipolar membrane electrodialysis (BMED) presents itself as an environmentally conscious and sustainable method for commercial implementation. This is achieved by the integration of the traditional electrodialysis process with water dissociation within a bipolar membrane. The process of converting salt into base and acid by BMED from wastewater with elevated salinity and organic content presents a novel approach to the recycling of raw materials and achieving zero liquid discharge [276,277].

### 4.5. Treatment of Dyes Using Hybrid Technologies

In recent times, there has been a growing interest in hybridized techniques. The importance of a process that combines elements from a blend process measure may be described as "synergistic" and "combinatorial" in nature. These solutions are characterized by their efficiency since they include the utilization of a single container to execute several tasks. The technique shown in Figure 11 is an integrated or blended approach for the treatment of dye waste. Furthermore, it is important to clearly articulate and acknowledge significant modifications in the advantages of hybrid methodologies. The subdivision of the hybrid technique, along with its associated benefits, is presented in Table S3.

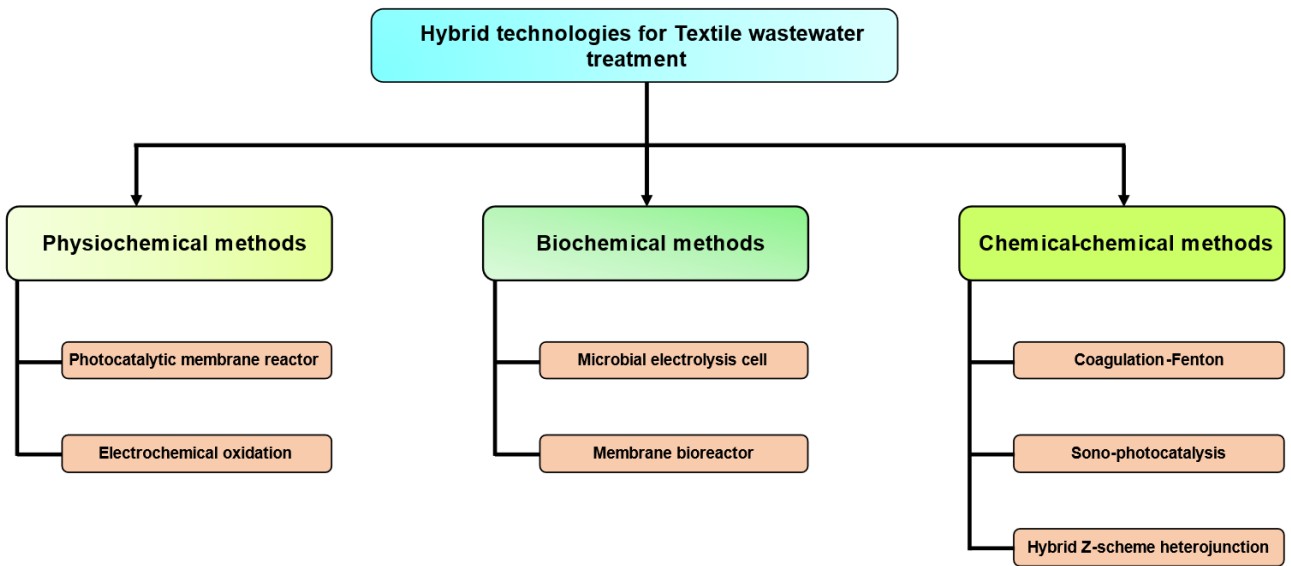

**Figure 11.** Treatment of dyes using hybrid technologies.

### 4.5.1. Physiochemical Methods

The addition of chemical-physical techniques is a further consideration to be considered in the removal of textile effluents. A photocatalytic/membrane separation (PMS) system consists of a photocatalyst, $TiO_2$, dispersed on a membrane, which is then placed in a photoreactor. Blended submerged membrane photoreactor (sMPR) frameworks exhibit superior photocatalytic removal efficiency compared to PMS due to the utilization of immobilized $TiO_2$ on the membrane surface. The PMS framework facilitates the separation of $TiO_2$ particles for the purpose of reuse. The hybrid process, known as PMS, utilizes a compact reactor and demonstrates energy efficiency while effectively eliminating complex and toxic pollutants [278,279]. This study investigates the photodegradation of Reactive Black 5 (RB5) in a slurry membrane reactor, examining both batch and continuous operational modes. The rate of color removal is higher at lower initial dye concentrations, and it increases as the concentration of ZnO increases up to 1.25 mg/L. The highest rate of RB5 removal, close to 100% within 60 min, was observed at pH 11 due to the combined effects of dye photolysis and the photoactivity of ZnO [280].

### 4.5.2. Biochemical Methods

Biological approaches represent costly and ecologically sustainable techniques for the removal of intricate azo dyes from waste effluents. However, their efficacy is limited when it comes to the elimination of various dye kinds. In contrast, it has been shown that aromatic amines, namely the molecules of dyes, exhibit a notable level of resistance against the process of biodegradation. The presence of degraded byproducts from azo dye effluents has been observed to impede cell motility and metabolic activity, hence hindering the efficacy of biological treatment approaches [281,282].

### 4.5.3. Combination-Based Hybrid Chemical–Chemical Scheme

In recent times, there has been a growing interest in the utilization of chemical methodologies in combination. Various AOPs, such as coagulation, utilization of Fenton chemicals, sono-photocatalysis, and the implementation of the photocatalytic hybrid Z-scheme, have demonstrated efficacy in the degradation of diverse hazardous and organic pollutants. Photocatalysis, specifically AOPs, facilitates the acceleration of a photoreaction by the absorption of photons from light. This absorption leads to the generation of electron ($e^-$)/hole ($h^+$) pairs, which actively engage in the redox reaction responsible for the degradation of pollutants [283–285].

### 4.5.4. The Z-Scheme Strategy

Despite the development of several wastewater treatment systems, there are still obstacles in the pursuit of efficient approaches to address pollution from complex waste effluents containing organic molecules and heavy metal ions, among other substances. The hybrid Z-scheme system is acquired by the careful selection of two semiconductors based on their suitable bandgap structure, as illustrated in Figure 12a. The design of highly effective Z-scheme heterojunction photocatalysts is a crucial and formidable task in the realm of wastewater purification utilizing solar energy. According to reports, organic pollutants such as dyes, phenol derivatives, and antibiotics can undergo effective degradation by oxidation by photogenerated holes ($h^+$), superoxide radicals ($O_2^-$), and hydroxyl radicals (OH) [286–290].

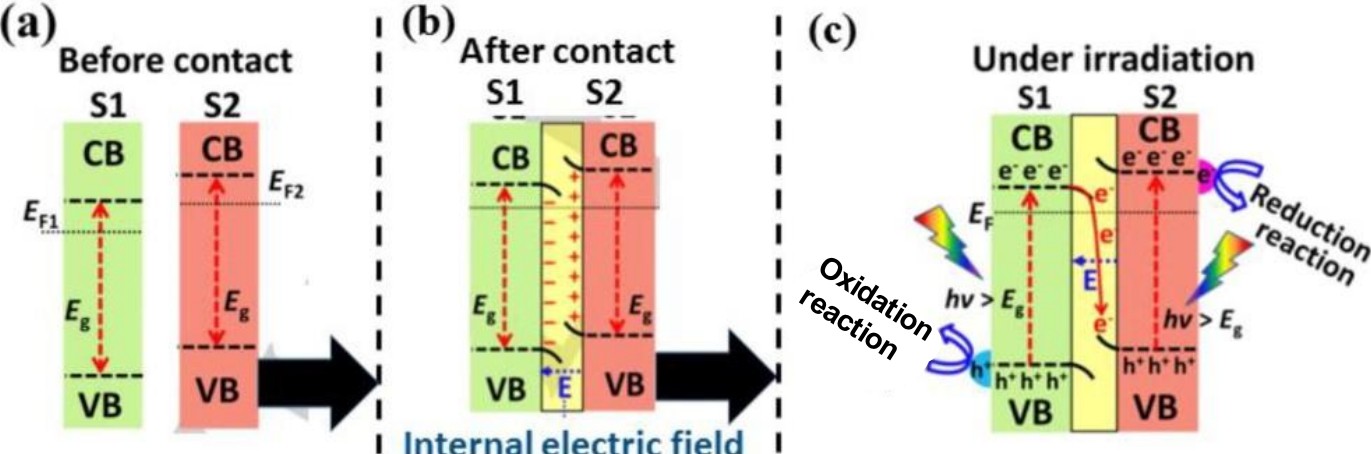

**Figure 12.** Schematic illustration of semiconductor heterojunction (**a**) before contact; (**b**) after contact; and (**c**) hybrid Z-scheme of photogenerated charge carriers (reprinted from [291]).

When two semiconductors are brought into contact, there is a phenomenon of charge separation that takes place at the interfaces. This separation is primarily caused by the difference in Fermi levels between the two semiconductors, as seen in Figure 12b. The semiconductor denoted as S1, which possesses a relatively low Fermi energy level, undergoes the acceptance of electrons from the semiconductor denoted as S2, which has a higher Fermi energy level. Consequently, an internal electric field is established at the interface, directing the flow of electrons from S2 to S1, as seen in Figure 12b. As depicted in Figure 12c, the interface exhibits an electric field that exerts a propulsive influence on the recombination of non-utilizable photogenerated electrons in the conduction band of S1 and non-utilizable photogenerated holes in the valence band of S2. Additionally, this electric field retains the electrons and holes, thereby enhancing their capacity to engage in the redox reaction within the conduction band of S2 and the valence band of S1, respectively. Consequently, these processes contribute significantly to the photocatalytic activity [291].

### 4.6. Sustainable Sludge Management

Sludge refers to the leftover, semi-solid substance that remains after the treatment of wastewater generated from textile processes. During the physical-chemical treatment process, the release of heavy metal concentration results in the formation of a chemical sludge. In contrast, impoverished soils might experience an additional advantage through the application of nutrient-rich biological sludge, which contains nitrogen and phosphorus, as well as useful organic matter. The primary issue is in the expeditious and forceful way sludge contaminates water sources. However, it is worth noting that certain locations within developing nations continue to dispose of sludge in inappropriate and environmentally unfriendly ways, such as through land disposal or by releasing it into the sea. To attain sustainable development, it is imperative to employ efficient recycling methods and utilize

waste materials properly, rather than resorting to burning or landfilling, which can result in the deposition of hazardous substances such as heavy metals. Practicing prudent management is crucial when it comes to the disposal of sludge, notwithstanding the inherent challenges involved. Various elements play a significant role in the management of sludge, including local and national geographical considerations, agronomic issues, economic aspects, and stakeholder perception [292,293].

Methods of Sludge Treatment

Anaerobic digestion refers to the process of decomposing sludge in an atmosphere devoid of oxygen. The significant characteristics of anaerobic digestion are the reduction in mass, formation of methane, and enhancement of dewatering qualities in the fermented sludge [292–297]. A higher level of investment in the digesting chamber is associated with a slower pace of deterioration. To enhance the biodegradability of sludge, several pre-treatment methods may be employed. These include thermal pre-treatment, enzymatic treatment, ozonation, chemical solubilization by acidification or alkaline hydrolysis, as well as mechanical sludge disintegration and ultrasonic pre-treatment [298,299]. The hypothesis posits that the process of anaerobic digestion of textile waste leads to the generation of biogas, as evidenced by the works [300–302]. This relationship is depicted in Figure 13.

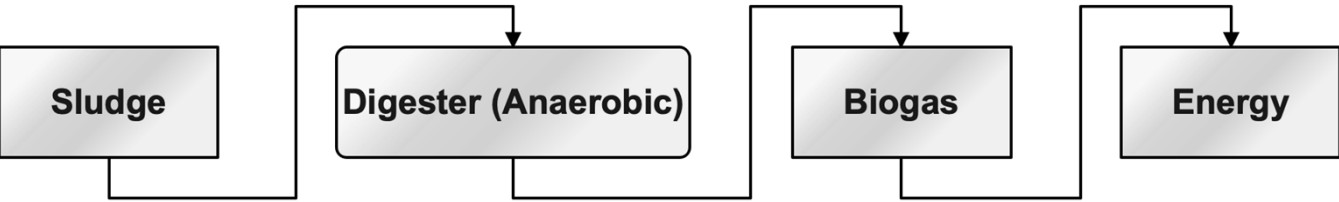

**Figure 13.** The sequence process for energy conversion from the sludge.

Aerobic digestion refers to the utilization of microorganisms within an oxygen-rich environment to facilitate the oxidation and decomposition of organic matter sludge. Aerobic sludge digestion is a procedure employed to decrease the levels of organic and inorganic constituents, as well as the overall volume, of sludge. The process under consideration exhibits temperature sensitivity and is susceptible to the presence of heavy metals, among other factors. However, it is noteworthy that despite its significant energy requirements, this process does not generate byproducts such as methane [303–305]. The pace of anaerobic digestion is constrained by the hydrolysis of organic materials in sludge, which subsequently leads to an increase in biogas output. This process serves as a pre-treatment technique that enhances the dewatering characteristics of the digested sludge [306]. The process of stabilizing sludge solids involves the application of several chemical treatments to the sludge in diverse manners. The application of polyelectrolytes as a conditioning agent for sludge dewatering operations has gained popularity due to its ability to enhance process yields [306–308].

*4.7. Roadmap towards ZLD: Focus on Recovery and Reuse*

Due to heightened environmental consciousness, escalating expenses related to wastewater treatment, and challenges pertaining to its disposal, there has been a perceptible shift in the public's perspective on wastewater. Presently, zero liquid discharge (ZLD) is emerging as a prospective preventative measure that plays a significant role in safeguarding the environment against the adverse impacts of industrial activity. The ZLD method in the textile industry focuses on achieving the goal of eliminating any disposal of liquid waste resulting from various waste-generating processes [94,309]. It is transitioning from being perceived as a nuisance that is conveniently ignored to being recognized as a potential avenue for the reclamation of precious resources. This phenomenon is manifesting as a direct consequence of the confluence of these variables. Figure 14 illustrates a graphical depiction of the fundamental factors that drive ZLD, along with its numerous beneficial results.

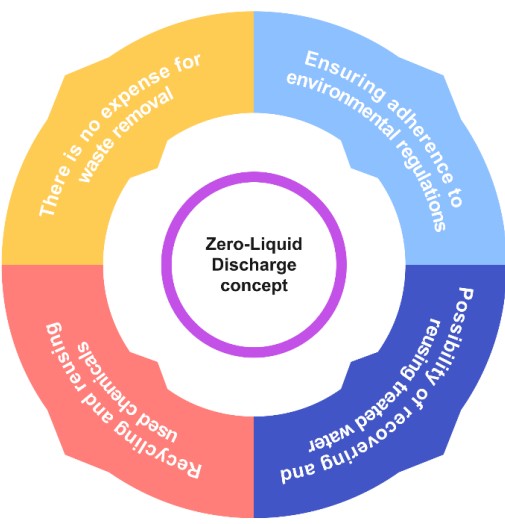

**Figure 14.** Benefits of the ZLD concept.

The practice of reusing and recycling wastewater has the potential to not only mitigate the demand for freshwater resources, but also facilitate the reduction in waste and surplus resources. The textile-processing sector is known for its significant use of colorants, chemicals, and other additives. Consequently, it has significant promise for several approaches to intense chemical recovery and water recycling. The process of recycling has become essential to the industrial sector due to the imposition of constraints on accessible water sources and regulations managing wastewater. Prior to commencing the treatment procedure, all potential avenues for recovery and recycling have been thoroughly explored and utilized to their fullest extent.

### 4.7.1. Electrolyte Recovery from Reactive Dye Effluent

Electrolytes are frequently employed in the dyeing process within the industry to help with the exhaustion of reactive and direct dyes. In some cases, the concentration of electrolytes employed might reach approximately 90 g/L, leading to substantial increases in the amounts of TDS and chlorides. These substances are known to display resistance to biodegradation. The dye bath and the first rinse bath are responsible for the liberation of approximately 80% of these salts [310]. In the given circumstances, empirical evidence has demonstrated that thermal evaporation treatment is the sole feasible alternative. Thermal evaporation is widely acknowledged as a very efficient technique employed in the wastewater treatment for the textile industry to removing salt and dissolved solids from concentrated effluent, as well as extraction of water. In the context of salt recovery, it is worth noting that the salts obtained following the evaporation process may be effectively employed in dyeing processes (i.e., same color), so contributing to a reduction in the utilization of virgin resources [311–313]; however, there are main drawback of this system can provide the mixture of colorants and salts. The partitioning of dye molecules and salts, particularly monovalent salts, can occur using a NF membrane. This approach effectively mitigates the release of detrimental substances into the surrounding ecosystem while concurrently reducing resource use, hence resulting in financial benefits [311,314,315].

### 4.7.2. Alkali Recovery

Mercerization is a significant textile finishing technique performed on cotton fabric, including the application of a concentrated sodium hydroxide solution (ranging from 20% to 30%) [316,317]. This procedure aims to enhance several characteristics of the fabric, such as its luster, tactile qualities, and other pertinent features. The hygroscopicity of the material is enhanced, resulting in increased strength and improved dye affinity. The recovery of NaOH is crucial when employing a high concentration of alkali. The reduction in effluent load and the recovery of NaOH is a very straightforward operation, as it does

not include the presence of additional chemicals, such as dyes. Membrane systems are considered very suitable for the treatment of effluents and the subsequent recovery of NaOH during the initial phases of fiber removal. Polymeric NF membranes allow for the passage of small molecules and ions without the need for mercerization. Nevertheless, several studies employ an ultrafiltration/microfiltration pre-treatment process due to the susceptibility of NF membranes to fouling, which might result in decreased penetration effectiveness [314,318–322]. The proposed approach involves a two-stage combination membrane design, incorporating a ceramic membrane in the first step and a polymeric membrane in the second phase. The recovery of sodium hydroxide from the procedure was found above 90%, recovered NaOH has the potential to be recycled into the process [318]. A separate investigation involved the utilization of microfiltration (MF) membranes with a pore size of 0.2 μm, followed by the implementation of UF membranes with a pore size of 100 kDa [319].

### 4.7.3. Dye Recovery

The effluent from dyebaths also results in the discharge of significant amounts of unabsorbed dyes, leading to a decrease in dissolved oxygen levels and the generation of high COD [71,323–325]. Moreover, the treatment of wastewater containing dyes poses a significant challenge in the field of wastewater management. Mariya et al. [326] aimed to assess the possibility for reusing colors obtained from denim and polyester dyebath effluents. In this investigation, tetraethylammonium bromide was employed as a draw solution. The findings of the study revealed that the forward osmosis membrane exhibited complete rejection of dyes, with a dye reconcentration ranging from 82% to 98%.

### 4.8. The Need for Technoeconomic Analysis

The discharge of effluent originating from the textile sector is resulting in substantial contamination due to the extensive use of colorants and hazardous chemicals, which are the primary contributors to water pollution and the escalating ecological risks. The escalating expenses associated with dye treatment facilities and the management of waste effluents are generating heightened public concern for environmental sustainability. The effective management and recovery of dyes and other chemical substances can contribute to the attainment of environmental sustainability and foster economic advancement. Furthermore, the substantial consumption of water underscores the imperative to engage in wastewater recycling. This has the potential to reduce the release of harmful compounds and create an atmosphere that is both safe and conducive to good health. The wastewaters under consideration consist of colors that include harmful and hazardous substances, such as pesticides, surfactants, and heavy metals. The cost of wastewater treatment often varies based on factors such as the specific treatment procedure employed, the spatial demands of the equipment, and the catalysts necessary for effectively removing contaminants. Furthermore, the operating expenses of the system encompass both personnel costs and maintenance expenditures. However, the prevailing price is contingent upon the region's authoritative needs, the availability of feedstock, and the labor force [291,327]. However, it should be noted that relying only on a single treatment procedure may not be a viable approach for effectively degrading extremely contaminated dyes. This is mostly due to the development of intermediates during the treatment, which afterwards necessitate an additional treatment step. Consequently, this supplementary treatment incurs additional costs, thereby impacting the overall feasibility of the process. Hence, the implementation of a plan that combines several approaches will be crucial in addressing efficiency. However, these procedures include the integration of two treatment approaches and offer greater advantages in terms of pollutant degradation.

### 4.9. Life Cycle Assessment (LCA) in WWTPs

A life cycle assessment (LCA) is a comprehensive approach used to evaluate the total environmental impact of a product or process across its entire life cycle, from raw material

extraction to disposal [328–331]. The primary goal of the LCA is to assess the whole range of environmental impacts resulting from the operation of the textile wastewater treatment facility. It is often essential to define the precise limits of the LCA investigation. The "Gate-to-Gate" methodology is a frequently used strategy in which different stages of the process are designated as gates. For example, we may classify the industry that emits pollutants as the first gate, and the release of cleaned effluent as the second gate. Furthermore, the analysis may also include the use of treated effluent in the textile sector. Moreover, it is crucial to establish a precise demarcation of the extent of the LCA studies carried out on wastewater treatment facilities (WWTPs). This encompasses delineating the parameters of the study, establishing the operational components, summarizing the LCA approach utilized, showcasing the life cycle inventory (LCI) information, and acknowledging the constraints, presumptions, and uncertainties linked to the examination.

LCA serves as a beneficial quantitative ecological assessment approach for examining various prospective functioning scenarios in the context of crucial water area planning. One of the advantages of LCA is in its ability to detect and quantify the impacts and influences of different process sequences, as well as assess the environmental effects of treatment technologies. LCA also facilitates the examination of pollution connections and aids in the achievement of effluent-free product creation. The LCA methodology was employed to conduct a comparative analysis of synthetic colors and natural colors, as well as synthetic finishes and biobased finishes, from an environmental perspective. Additionally, the study investigated the potential impacts of these materials on WWTPs. LCA offers decision makers and policy makers a consistent and transparent means to understand and interpret the ecological performance data of WWTPs in the textile sector. Hence, the utilization of LCA may facilitate the identification of research and development goals and provide guidance for enhancing innovation by mitigating challenges related to waste disposal and the discharge of hazardous chemicals. Figure 15 depicts the typical LCA framework on the WWTPs for the textile industry.

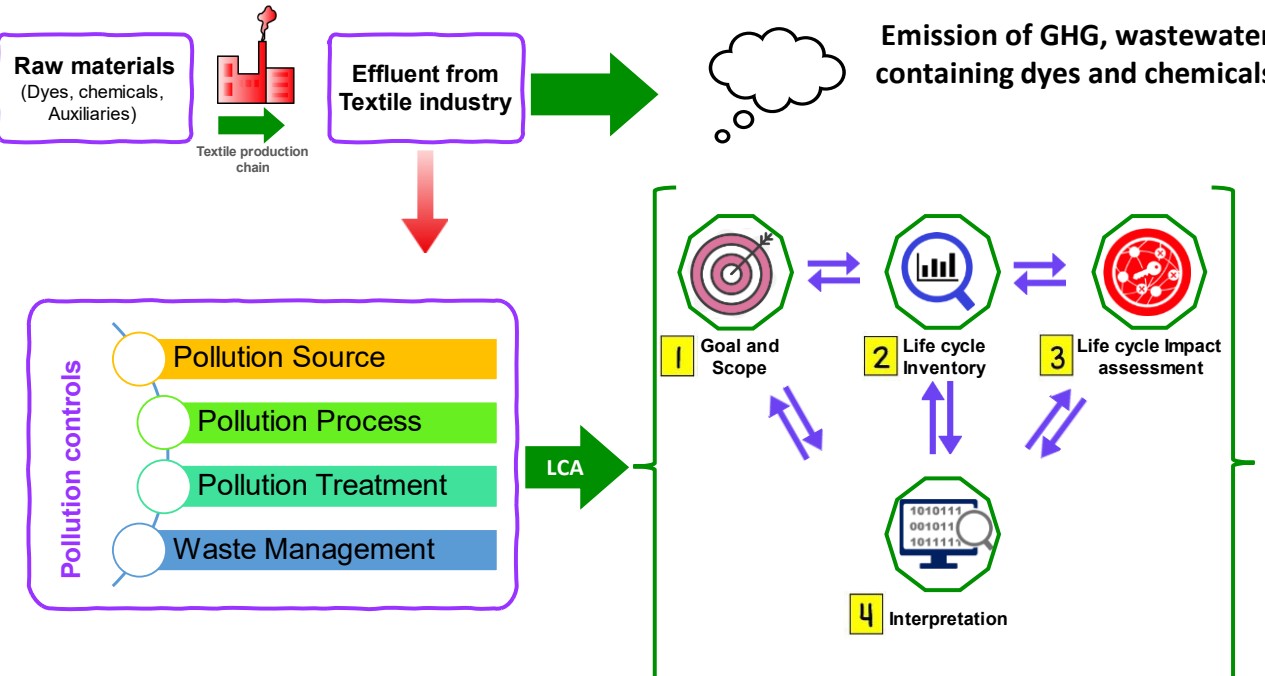

**Figure 15.** Life cycle assessment (LCA) framework for the wastewater treatment plants.

## 5. Conclusions

The textile industry is responsible for generating effluents that contain significant quantities of hazardous and persistent compounds, including dyes, chemicals, aromatic compounds, formaldehyde, flame retardants, and fluorocarbons. These substances have

detrimental impacts on both the environment and human health, as well as on aquatic organisms. This review paper demonstrated the comprehensive literature analysis of several techniques for sustainable treatment of textile wastewater, with a specific focus on bioadsorbents, biological approaches, membrane technology, ion exchange, advanced oxidation processes, as well as physicochemical and biochemical processes. The tabulation of different bioadsorbents and their respective adsorbent capacities has been conducted, including a discussion on energy-efficient and cost-effective membranes and other treatment systems. Furthermore, this paper focused on strategies that result in zero liquid discharge (ZLD) and the subsequent retrieval of significant resources, such as dyes, alkalis, and electrolytes, which are produced in huge quantities during wastewater treatment plants. In recent times, there has been a growing interest in the recovery of dyes from textile waste. Consequently, dye recovery has emerged as a significant subject within the context of textile circular economy and industrial urban symbiosis. Furthermore, this review paper analyzes the management of sludge, conducts a technoeconomic analysis, and highlights their significance within the textile value chain. It is important to perform a life cycle analysis of wastewater treatment plants to ensure the efficient management of their reuse, reduction, and disposal processes.

In this situation, the idea of prevention being more effective than treatments remains valid. The use of biomaterials for functionalization, such as coloring (i.e., mass colorations) and surface changes, has been widely investigated in the literature [73,332–335]. By incorporating more biobased fibers, including natural and regenerated fibers, as well as other biobased and biodegradable polymers, it is possible to mitigate the overall pollution burden on wastewater treatment plants. Furthermore, the application of mass coloring or mass functionalization in the process of melt extrusion, namely in the case of polylactic acid, or in manmade fibers like viscose, infinna, spinnova, renewcell and ioncel has the potential to decrease the pollution burden associated with both synthetic and regenerate fibers [336–339].

The textile industry uses effluent treatment to separate water and other substances, with water recovery being easier than salt or residues. Highly polluting effluents, such as dye bath discharge, make up 10% of total effluent discharge, while 90% comes from low-polluting streams like wash water. Sustainable wastewater treatment can be applied to both streams but requires careful selection of highly polluting effluent streams. Rejecting reverse osmosis, nano- and ultrafiltration, the advanced oxidation process and granular activated carbon should be used to treat highly polluting effluent streams. Sustainable wastewater treatment produces less sludge, but it does not comply with norms and standards. In certain cases, using multiple methods could improve pollutant removal efficiency.

Sustainable wastewater treatments have several advantageous attributes, indicating their preference for minimizing chemical use in wastewater treatment plants, reducing energy consumption, and mitigating the carbon impact, among other benefits. The promotion of initiatives aimed at showcasing sustainable wastewater treatments at the industrial level is vital to mitigate the environmental impact produced by wastewater treatment plants.

**Supplementary Materials:** The following supporting information can be downloaded at: https://www.mdpi.com/article/10.3390/su16020495/s1, Table S1. Advantages and disadvantages of biological methods for dye removal; Table S2. Dye degradation by various bacterial biomass; Table S3. Advantages of combination-based (hybrid) processes; Figure S1. Schematic representation of the forward osmosis process [47,59,66,340–351].

**Funding:** This research received no external funding.

**Institutional Review Board Statement:** Not applicable.

**Informed Consent Statement:** Not applicable.

**Data Availability Statement:** Not applicable.

**Conflicts of Interest:** The author declares no conflict of interest.

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
