# Peer review of "Recent Advances in the Remediation of Textile-Dye-Containing Wastewater: Prioritizing Human Health and Sustainable Wastewater Treatment"

_sustainability, doi:10.3390/su16020495_

Round 1
Reviewer 1 Report
Comments and Suggestions for Authors 1. What is the main question addressed by the research?This paper mainly summarized the environmental impacts and technical remediation of dye-containing wastewater, which is helpful for the primary audience to understand this field.
2. Do you consider the topic original or relevant in the field? Does it address a specific gap in the field?
There are some similar review papers was published, while this work was mainly focus on the comparable treatment in detailed.
3. What does it add to the subject area compared with other published material?
Same as question 2.
4. What specific improvements should the authors consider regarding the methodology? What further controls should be considered?
The non-toxic dyes are expected to synthesize in conjunction with these remediation treatments to effectively remove the dye-related pollution.
5. Are the conclusions consistent with the evidence and arguments presented and do they address the main question posed? Yes.
6. Are the references appropriate? Yes.
7. Please include any additional comments on the tables and figures. NO further comments.
In this review, the authors firstly presented the classification of textile dyes and their common treatment methods. According to the Human and environmental issues from effluents discharging, they mainly highlighted the sustainable approaches for the management wastewater in the textile industry. The manuscript is well organized and summarized to be helpful for readers to understand the current state and potential applications of wastewater treatment plants. Thus, I would like to recommend the acceptance of this manuscript after a minor revision. The following issues should be addressed:
(1) The uniform references in Supporting information should be corrected.
(2) In page of 4 (line 132), there may be some mistake in this sentence “ biological, chemical, physical, and various combinations thereof.”
Comments on the Quality of English LanguageMinor editing of English language required
Author Response
Dear reviwer
Thank you for spending the time to read my work and providing constructive and insightful suggestions. Your feedback has greatly improved the quality of the text. I have included a point-by-point respond to which I believe will be satisfactory to you.
Yours
Author

Reviewer 2 Report
Comments and Suggestions for Authors
The paper titled "Recent Advances in the Remediation of Textile Dye-containing Wastewater: Prioritizing Human Health and Sustainable Wastewater Treatment" offers a comprehensive review of the subject matter pertaining to wastewater containing textile dyes, with a particular emphasis on its impact on human health. The paper also highlights the significance of sustainable wastewater treatment, which is currently a prominent area of interest within the fields of sustainability and environmental science. In general, the work appears to be of acceptable quality. I would propose minor revisions to address a few issues that need to be resolved before publishing.
· Abbreviation is not provided properly, this needs to be checked for the entire paper.
· Need some more description on the nanofiltration, is it available commercially?
· The author is required to present a rationale for the significance of Life Cycle Assessment (LCA) in the context of Wastewater Treatment Plants (WWTPs).
· The author may consider augmenting the material pertaining to bioadsorbents.
· The author is required to undertake a revision of the template utilized in the parts pertaining to acknowledgments and contributions.
· Some typo errors are there for instance Fig 15.
One of the point that was not covered in the manuscipt, is the degradation of dyes, degraded metabolites, and their impact on the envrionment or humans. However, it is a suromounted task to cover each aspect of this field of study as it is quite diverse and requires expertise, therefore it is suggestion to author that such study may consider in future publication, seeing lesser publication in this subcategory of this topic.
As a general application of this topic, it is apparent that the article's material exhibits an adequate level of structure and possesses the potential to get heightened citations in subsequent academic publications.
Author Response

(The authors gave the same response as above.)

Reviewer 3 Report
Comments and Suggestions for Authors
This is a very good topic of the review paper (Manuscript Number: sustainability-2748504), the author summarized the research results on the remediation of industrial wastewater in the textile industry. The effects of textile wastewater containing dyes and chemicals were evaluated comprehensively, and the potential effects on human health, aquatic health, and the environment were discussed. In addition, an in-depth review of various environmentally sustainable approaches to the management and treatment of wastewater from the textile industry was conducted. The importance of recycling dyes, alkalis, and 23 electrolytes in wastewater treatment was emphasized. It is a carefully done study and the findings are of considerable interest. However, I thought it still has some deficiencies and I recommend revisions before publication.
1) In my opinion, the novelty of this work is questionable. This paper summarizes the research results of textile industry wastewater treatment, which is a very simple work, but its scope is very broad, and it is of great help to the research community. This is worth studying. The author uses several words repeatedly throughout the article. There are a few sentences written awkwardly. I suggest that this article be carefully revised in English.
2) The picture quality is too poor. Some pictures do not meet the requirements of scientific research papers. In Figure 1, is it appropriate to present this kind of full-text content as a picture? In Figure 5, should the fonts used in the same picture be consistent? In Figure 8 and Figure 13, is a colorful background necessary? Should the font in Figure 14 be properly enlarged? The sharpness of some Figures should be improved. As shown in Figure 12, the picture is somewhat vague. Figure 5 (d) is unclear. Please consider these questions carefully.
3) The subgraphs (a) and (b) in Figure 7 are easy to confuse the reader, so it is suggested that the author use a clearer division of the subgraphs (a) and (b). You can add a separate picture box as in Figure 9 and Figure 10. Consistent format
4) The logic of the thesis is clear, but the length of Section 4 "Sustainable wastewater treatment for the remediation" is confusing, especially the length of Section 4.3 is too long. Section 4.3 is "Other techniques", but the actual narrative length is much longer than Section 4.1 and Section 4.2, is there any mistake in content arrangement?
5) Throughout the whole paper, the author makes a detailed summary of the research results in this research field, but hopes that the author can form his own new opinions and conclusions, rather than simply listing the previous research results.
6) The author lacks basic skills and qualities in writing scientific research papers. The background of "NEED" in the title of Figure 15 is even marked in green, you should not leave traces of revision marks in the final paper, which goes against the rigor of the paper writing
Author Response

(The authors gave the same response as above.)

Reviewer 4 Report
Comments and Suggestions for Authors
Dear Author,
The manuscript submitted for review presents an overview of various techniques for the treatment of wastewater from the textile industry. This work contains a great number of topics and makes a kind of compendium of specific knowledge. It should be noted that the section devoted to the use, during the neutralisation of textile wastewater waste of natural origin (for example, coconut dust, citrus lime peels, or tea leaf waste). It can be noted that efforts have been made to indicate what the circular economy of this kind of wastewater could look like.
1. The first part of the theoretical introduction needs to be rethinking. The sentences are not logically connected to each other. There is no ordering of the content, and numerous errors of a grammatical nature make it even more difficult to follow the author's thought. This remark applies mainly to the paragraphs on lines 31 to 53. Similarly, the matter concerns the summary, which is supposed to highlight the most important things. In this case, it is written to the point. The author suggests that a techno-economic analysis of the issue in question has been made, while this is not the case. The article deals primarily with ways to neutralise the textile industry. It does not say much about the economic aspects. Therefore, these issues need to be supplemented and given more commentary and thought about what is worth emphasising and highlighting.
2. In Figure 5, in parts (a) and (d), important details are not visible, they are too small and blurred. What is more puzzling is the third-order reaction shown in part (a). Either it needs an addition or some special commentary, as it is wrong in this form.
3. Part of Figure 9(b) is illegible. Therefore, the assessment of this photo cannot be made. If the photo contains relevant information for the manuscript, then it should be of adequate quality.
4. There is a lot of confusion in photo 10 and in the description assigned to it. How is it possible that iron compounds are blue in colour? Does the reaction run in the direction shown by the author or in the opposite direction. Standard potential values for partial redox reactions show that the reaction occurs in the direction of oxidation of Fe2+ to Fe3+ (abstracting from the fact that clearly some copper compounds and not iron is shown in the photos). Moreover, the charges of the OH- ions and other functional groups are missing or invisible in the photo.
5. I am also curious about the source from which the sentences from lines 669-673 were written. If we assume that oxygen is formed in a reaction in which one of the reactants is hydrogen peroxide, then this happens in an acidic environment and not an alkaline one. In addition, anywhere in the publication, where a hydroxyl group appears, it should be spelt with proper care, i.e. OH-.
I hope that my comments will help to improve the content value of this publication.
Sincerely Yours,
Reviewer
Comments on the Quality of English LanguageDear Author,
Analysing the submitted manuscript, one can see quite a lot of linguistic and grammatical errors. It is puzzling that most of them are in the two most relevant parts of the publication, namely, the theoretical introduction and the conclusions. In this case, I think it is worth showing the text to two people after the corrections. One, who is unfamiliar with the subject but knows English - just to see if it is understandable and logical. The other is to ensure that grammatical errors have been avoided.
Sincerely Yours,
Reviewer
Author Response

(The authors gave the same response as above.)

Round 2
Reviewer 4 Report
Comments and Suggestions for Authors
Dear Author,
The manuscript submitted for review in the current version has much greater merit. Key passages have been rewritten and unfortunate wording removed and replaced with more thoughtful text. The description of Figures 5 and 10 is not controversial. Although nothing can be done with the illustrations themselves since they are from other publications. However, what could be improved has been done and can remain as is.
Sincerely Yours,
Reviewer
Comments on the Quality of English LanguageDear Author,
In key sections of the manuscript, the quality of the statements has been improved. In the earlier version, it was not the English language that was the biggest problem, but imprecise wording and sentences that did not form a coherent plot. At this point, the manuscript reads very well. There is still a problem of misunderstood commas or the absence of commas and missing articles. However, this does not change my positive assessment of the text from a linguistic point of view.
Sincerely Yours,
Reviewer